# UNDERSTANDING THE FAILURE MODES OF OUT-OF-DISTRIBUTION GENERALIZATION

**Vaishnavh Nagarajan**[*]
Carnegie Mellon University
vaishnavh@cs.cmu.edu

**Anders Andreassen**
Blueshift, Alphabet
ajandreassen@google.com

**Behnam Neyshabur**
Blueshift, Alphabet
neyshabur@google.com

## ABSTRACT

Empirical studies suggest that machine learning models often rely on features, such as the background, that may be spuriously correlated with the label only during training time, resulting in poor accuracy during test-time. In this work, we identify the fundamental factors that give rise to this behavior, by explaining why models fail this way *even* in easy-to-learn tasks where one would expect these models to succeed. In particular, through a theoretical study of gradient-descent-trained linear classifiers on some easy-to-learn tasks, we uncover two complementary failure modes. These modes arise from how spurious correlations induce two kinds of skews in the data: one *geometric* in nature, and another, *statistical* in nature. Finally, we construct natural modifications of image classification datasets to understand when these failure modes can arise in practice. We also design experiments to isolate the two failure modes when training modern neural networks on these datasets.[1]

## 1 INTRODUCTION

A machine learning model in the wild (e.g., a self-driving car) must be prepared to make sense of its surroundings in rare conditions that may not have been well-represented in its training set. This could range from conditions such as mild glitches in the camera to strange weather conditions. This out-of-distribution (OoD) generalization problem has been extensively studied within the framework of the domain generalization setting (Blanchard et al., 2011; Muandet et al., 2013). Here, the classifier has access to training data sourced from multiple "domains" or distributions, but no data from test domains. By observing the various kinds of shifts exhibited by the training domains, we want the classifier can learn to be robust to such shifts.

The simplest approach to domain generalization is based on the Empirical Risk Minimization (ERM) principle (Vapnik, 1998): pool the data from all the training domains (ignoring the "domain label" on each point) and train a classifier by gradient descent to minimize the average loss on this pooled dataset. Alternatively, many recent studies (Ganin et al., 2016; Arjovsky et al., 2019; Sagawa et al., 2020a) have focused on designing more sophisticated algorithms that do utilize the domain label on the datapoints e.g., by enforcing certain representational invariances across domains.

A basic premise behind pursuing such sophisticated techniques, as emphasized by Arjovsky et al. (2019), is the empirical observation that ERM-based gradient-descent-training (or for convenience, just ERM) fails in a characteristic way. As a standard illustration, consider a cow-camel classification task (Beery et al., 2018) where the background happens to be spuriously correlated with the label in a particular manner only during training — say, most cows are found against a grassy background and most camels against a sandy one. Then, during test-time, if the correlation is completely flipped (i.e., *all* cows in deserts, and *all* camels in meadows), one would observe that the

---

[*]Work performed in part while Vaishnavh Nagarajan was interning at Blueshift, Alphabet.
[1]Code is available at https://github.com/google-research/OOD-failures

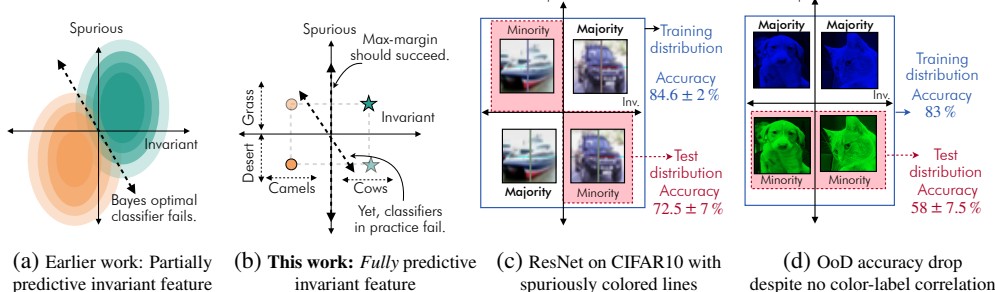

(a) Earlier work: Partially predictive invariant feature

(b) **This work:** *Fully* predictive invariant feature

(c) ResNet on CIFAR10 with spuriously colored lines

(d) OoD accuracy drop despite no color-label correlation

Figure 1: **Unexplained OoD failure**: Existing theory can explain why classifiers rely on the spurious feature when the invariant feature is in itself not informative enough (**Fig 1a**). But when invariant features are fully predictive of the label, these explanations fall apart. E.g., in the four-point-dataset of **Fig 1b**, one would expect the max-margin classifier to easily ignore spurious correlations (also see Sec 3). Yet, why do classifiers (including the max-margin) rely on the spurious feature, in so many real-world settings where the shapes are perfectly informative of the object label (e.g., **Fig 1c**)? We identify two fundamental factors behind this behavior. In doing so, we also identify and explain other kinds of vulnerabilities such as the one in **Fig 1d** (see Sec 4).

accuracy of ERM drops drastically. Evidently, ERM, in its unrestrained attempt at fitting the data, indiscriminately relies on all kinds of informative features, including unreliable *spurious features* like the background. However, an algorithm that carefully uses domain label information can hope to identify and rely purely on *invariant features* (or "core" features (Sagawa et al., 2020b)).

While the above narrative is an oft-stated motivation behind developing sophisticated OoD generalization algorithms, there is little formal explanation as to *why* ERM fails in this characteristic way. Existing works (Sagawa et al., 2020b; Tsipras et al., 2019; Arjovsky et al., 2019; Shah et al., 2020) provide valuable answers to this question through concrete theoretical examples; however, their examples critically rely on certain factors to make the task difficult enough for ERM to rely on the spurious features. For instance, many of these examples have invariant features that are only partially predictive of the label (see Fig 1a). Surprisingly though, ERM relies on spurious features even in much easier-to-learn tasks where these complicating factors are absent — such as in tasks with fully predictive invariant features e.g., Fig 1c or the Waterbirds/CelebA examples in Sagawa et al. (2020a) or for that matter, in any real-world situation where the object shape perfectly determines the label. This failure in easy-to-learn tasks, as we argue later, is not straightforward to explain (see Fig 1b for brief idea). This evidently implies that there must exist factors more general and fundamental than those known so far, that cause ERM to fail.

Our goal in this work is to uncover these fundamental factors behind the failure of ERM. The hope is that this will provide a vital foundation for future work to reason about OoD generalization. Indeed, recent empirical work (Gulrajani & Lopez-Paz, 2020) has questioned whether existing alternatives necessarily outperform ERM on OoD tasks; however, due to a lack of theory, it is not clear how to hypothesize about when/why one algorithm would outperform another here. Through our theoretical study, future work can hope to be better positioned to precisely identify the key missing components in these algorithms, and bridge these gaps to better solve the OoD generalization problem.

**Our contributions.** To identify the most fundamental factors causing OoD failure, our strategy is to (a) study tasks that are "easy" to succeed at, and (b) to demonstrate that ERM relies on spurious features *despite* how easy the tasks are. More concretely:

1. We formulate a set of constraints on how our tasks must be designed so that they are easy to succeed at (e.g., the invariant feature must be fully predictive of the label). Notably, this class of *easy-to-learn tasks* provides both a theoretical test-bed for reasoning about OoD generalization and also a simplified empirical test-bed. In particular, this class encompasses simplified MNIST and CIFAR10-based classification tasks where we establish empirical failure of ERM.

2. We identify two complementary mechanisms of failure of ERM that arise from how spurious correlations induce two kinds of skews in the data: one that is *geometric* and the other *statistical*. In particular, we theoretically isolate these failure modes by studying linear classifiers trained by

gradient descent (on logistic/exponential loss) and its infinite-time-trained equivalent, the max-margin classifier (Soudry et al., 2018; Ji & Telgarsky, 2018) on the easy-to-learn-tasks.

3. We also show that in any easy-to-learn task that does not have these geometric or statistical skews, these models do not rely on the spurious features. This suggests that these skews are not only a sufficient but also a necessary factor for failure of these models in easy-to-learn tasks.

4. To empirically demonstrate the generality of our theoretical insights, we (a) experimentally validate these skews in a range of MNIST and CIFAR10-based tasks and (b) demonstrate their effects on fully-connected networks (FNNs) and ResNets. We also identify and explain failure in scenarios where standard notions of spurious correlations do not apply (see Fig 1d). We perform similar experiments on a non-image classification task in App E.

## 2 RELATED WORK

**Spurious correlations.** Empirical work has shown that deep networks find superficial ways to predict the label, such as by relying on the background (Beery et al., 2018; Ribeiro et al., 2016) or other kinds of shortcuts (McCoy et al., 2019; Geirhos et al., 2020). Such behavior is of practical concern because accuracy can deteriorate under shifts in those features (Rosenfeld et al., 2018; Hendrycks & Dietterich, 2019). It can also lead to unfair biases and poor performance on minority groups (Dixon et al., 2018; Zhao et al., 2017; Sagawa et al., 2020b).

**Understanding failure of ERM.** While the fact that ERM relies on spurious correlations has become empirical folk wisdom, only a few studies have made efforts to carefully model this. Broadly, there are two kinds of existing models that explain this phenomenon. One existing model is to imagine that both the invariant and the spurious features are only partially predictive of the label (Tsipras et al., 2019; Sagawa et al., 2020b; Arjovsky et al., 2019; Ilyas et al., 2019; Khani & Liang, 2020), as a result of which the classifier that maximizes accuracy cannot ignore the spurious feature (see Fig 1a). The other existing model is based on the "simplicity bias" of gradient-descent based deep network training (Rahaman et al., 2018; Neyshabur et al., 2015; Kalimeris et al., 2019; Arpit et al., 2017; Xu et al., 2019; des Combes et al., 2018). In particular, this model typically assumes that both the invariant and spurious features are fully predictive of the label, but crucially posits that the spurious features are simpler to learn (e.g., more linear) than the invariant features, and therefore gradient descent prefers to use them (Shah et al., 2020; Nar et al., 2019; Hermann & Lampinen, 2020).

While both these models offer simple-to-understand and useful explanations for why classifiers may use spurious correlations, we provide a more fundamental explanation. In particular, we empirically and theoretically demonstrate how ERM can rely on the spurious feature even in much easier tasks where these explanations would fall apart: these are tasks where unlike in the first model (a) the invariant feature is fully predictive *and* unlike in the second model, (b) the invariant feature corresponds to a simple linear boundary *and* (c) the spurious feature is not fully predictive of the label. Further, we go beyond the max-margin settings analyzed in these works to analyze the dynamics of finite-time gradient-descent trained classifier on logistic loss. We would also like to point the reader to concurrent work of Khani & Liang (2021) that has proposed a different model addressing the above points. While their model sheds insight into the role of overparameterization in the context of spurious features (and our results are agnostic to that), their model also requires the spurious feature to be "dependent" on the invariant feature, an assumption we don't require (see Sec 3).

**Algorithms for OoD generalization.** Due to the empirical shortcomings of ERM, a wide range of sophisticated algorithms have been developed for domain generalization. The most popular strategy is to learn useful features while constraining them to have similar distributions across domains (Ganin et al., 2016; Li et al., 2018b; Albuquerque et al., 2020). Other works constrain these features in a way that one can learn a classifier that is simultaneously optimal across all domains (Peters et al., 2016; Arjovsky et al., 2019; Krueger et al., 2020). As discussed in Gulrajani & Lopez-Paz (2020), there are also many other existing non-ERM based methods, including that of meta-learning (Li et al., 2018a), parameter-sharing (Sagawa et al., 2020a) and data augmentation (Zhang et al., 2018). Through their extensive empirical survey of many of the above algorithms, Gulrajani & Lopez-Paz (2020) suggest that ERM may be just as competitive as the state-of-the-art. But we must emphasize that this doesn't vindicate ERM of its failures but rather indicates that we may be yet to develop a substantial improvement over ERM.

## 3 EASY-TO-LEARN DOMAIN GENERALIZATION TASKS

Below, we first set up the basic domain generalization setting and the idea of ERM. Then, in Section 3.1, we formulate a class of domain-generalization tasks that are in many aspects "easy" for the learner (such as fully informative invariant features) – what exactly makes a task "easy" will be discussed in Section 3.1). This discussion sets the ground for the later sections to show how ERM can fail even in these easy tasks, which will help uncover the fundamental factors behind its failure.

**Notations.** Consider an input (vector) space $\mathcal{X}$ and label space $\mathcal{Y} \in \{-1, 1\}$. For any distribution $\mathcal{D}$ over $\mathcal{X} \times \mathcal{Y}$, let $p_{\mathcal{D}}(\cdot)$ denote its probability density function (PDF). Let $\mathbb{H}$ denote a class of classifiers $h : \mathcal{X} \to \mathbb{R}$. Let the error of $h$ on $\mathcal{D}$ be denoted as $L_{\mathcal{D}}(h) := \mathbb{E}_{(\mathbf{x},y)\sim\mathcal{D}}[h(\mathbf{x}) \cdot y < 0]$.

**The domain generalization setting and ERM.** In the domain generalization setting, one considers an underlying class $\mathbb{D}$ of data distributions over $\mathcal{X} \times \mathcal{Y}$ corresponding to different possible domains. The learner is given training data collected from multiple distributions from $\mathbb{D}$. For an ERM-based learner in particular, the training data will be pooled together, so we can model the data as coming from a single (pooled) distribution $\mathcal{D}_{\text{train}}$, which for simplicity, can be assumed to belong to $\mathbb{D}$. Given this data, the learner outputs a hypothesis $\hat{h} \in \mathbb{H}$ that is tested on a new distribution $\mathcal{D}_{\text{test}}$ picked from $\mathbb{D}$. This can be potentially modeled by assuming that all test and training distributions are drawn from a common hyper-distribution over $\mathbb{D}$. However, this assumption becomes pointless in most practical settings where the training domains are not more than three to four in number (e.g., PACS (Asadi et al., 2019), VLCS (Fang et al., 2013)), and therefore hardly representative of any hyper-distribution. Here, the problem becomes as hard as ensuring good performance on a worst-case test-distribution without any hyper-distribution assumption; this boils down to minimizing $\max_{\mathcal{D}\in\mathbb{D}} L_{\mathcal{D}}(\hat{h})$. Indeed, most works have studied the worst-case setting, both theoretically (Sagawa et al., 2020b) and empirically (Arjovsky et al., 2019; Sagawa et al., 2020a).

Similarly, for this work, we focus on the worst-case setting and define the optimal target function $h^{\star}$ to be $h^{\star} = \arg\min_{h\in\mathbb{H}} \max_{\mathcal{D}\in\mathbb{D}} L_{\mathcal{D}}(h)$. Then, we define the features that this "robust" classifier uses as invariant features $\mathcal{X}_{\text{inv}}$ (e.g., the shape of the object), and the rest as spurious features $\mathcal{X}_{\text{sp}}$ (e.g., the background). To formalize this, we assume that there exists a mapping $\Phi : \mathcal{X}_{\text{inv}} \times \mathcal{X}_{\text{sp}} \to \mathcal{X}$ such that each $\mathcal{D} \in \mathbb{D}$ is induced by a distribution over $\mathcal{X}_{\text{inv}} \times \mathcal{X}_{\text{sp}}$ (so we can denote any $\mathbf{x}$ as $\Phi(\mathbf{x}_{\text{inv}}, \mathbf{x}_{\text{sp}})$). With an abuse of notation we will use $p_{\mathcal{D}}(\cdot)$ to also denote the PDF of the distribution over $\mathcal{X}_{\text{inv}} \times \mathcal{X}_{\text{sp}}$. Then, the fact that $\mathcal{X}_{\text{sp}}$ are features that $h^{\star}$ does not rely on, is mathematically stated as: $\forall \mathbf{x}_{\text{inv}}$ and $\forall \mathbf{x}_{\text{sp}} \neq \mathbf{x}'_{\text{sp}}$, $h^{\star}(\Phi(\mathbf{x}_{\text{inv}}, \mathbf{x}_{\text{sp}})) = h^{\star}(\Phi(\mathbf{x}_{\text{inv}}, \mathbf{x}'_{\text{sp}}))$. Finally, we note that, to make this learning problem tractable, one has to impose further restrictions; we'll provide more details on those when we discuss the class of easy-to-learn domain generalization tasks in Sec 3.1.

**Empirical failure of ERM.** To guide us in constructing the easy-to-learn tasks, let us ground our study in a concrete empirical setup where an ERM-based linear classifier shows OoD failure. Specifically, consider the following Binary-MNIST based task, where the first five digits and the remaining five digits form the two classes. First, we let $\Phi$ be the identity mapping, and so $\mathbf{x} = (\mathbf{x}_{\text{inv}}, \mathbf{x}_{\text{sp}})$. Then, we let $\mathbf{x}_{\text{inv}}$ be a random ReLU features representation of the MNIST digit i.e., if $\mathbf{x}_{\text{raw}}$ represents the MNIST image, then $\mathbf{x}_{\text{inv}} = \text{ReLU}(W\mathbf{x}_{\text{raw}})$ where $W$ is a matrix with Gaussian entries. We make this representation sufficiently high-dimensional so that the data becomes linearly separable. Next, we let the spurious feature take values in $\{+\mathcal{B}, -\mathcal{B}\}$ for some $\mathcal{B} > 0$, imitating the two possible background colors in the camel-cow dataset. Finally, on $\mathcal{D}_{\text{train}}$, for any $y$, we pick the image $\mathbf{x}_{\text{inv}}$ from the corresponding class and independently set the "background color" $x_{\text{sp}}$ so that there is some spurious correlation i.e., $\text{Pr}_{\mathcal{D}_{\text{train}}}[x_{\text{sp}} \cdot y > 0] > 0.5$. During test time however, we flip this correlation around so that $\text{Pr}_{\mathcal{D}_{\text{test}}}[x_{\text{sp}} \cdot y > 0] = 0.0$. In this task, we observe in Fig 2a (shown later under Sec 4) that as we vary the train-time spurious correlation from none ($\text{Pr}_{\mathcal{D}_{\text{train}}}[x_{\text{sp}} \cdot y > 0] = 0.5$) to its maximum ($\text{Pr}_{\mathcal{D}_{\text{train}}}[x_{\text{sp}} \cdot y > 0] = 1.0$), the OoD accuracy of a max-margin classifier progressively deteriorates. (We present similar results for a CIFAR10 setting, and all experiment details in App C.1.) Our goal is now to theoretically demonstrate why ERM fails this way (or equivalently, why it relies on the spurious feature) even in tasks as "easy-to-learn" as these.

### 3.1 CONSTRAINTS DEFINING EASY-TO-LEARN DOMAIN-GENERALIZATION TASKS.

To formulate *a class of easy-to-learn tasks*, we enumerate a set of constraints that the tasks must satisfy; notably, this class of tasks will encompass the empirical example described above. The

motivation behind this exercise is that restricting ourselves to the constrained set of tasks yields stronger insights — it prevents us from designing complex examples where ERM is forced to rely on spurious features due to a not-so-fundamental factor. Indeed, each constraint here forbids a unique, less fundamental failure mode of ERM from occuring in the easy-to-learn tasks.

**Constraint 1. (Fully predictive invariant features.)** For all $\mathcal{D} \in \mathbb{D}$, $L_\mathcal{D}(h^\star) = 0$.
Arguably, our most important constraint is that the invariant features (which is what $h^\star$ purely relies on) are *perfectly informative* of the label. The motivation is that, when this is not the case (e.g., as in noisy invariant features like Fig 1a or in Sagawa et al. (2020b); Tsipras et al. (2019)), failure can arise from the fact that the spurious features provide vital extra information that the invariant features cannot provide (see App A for more formal argument). However this explanation quickly falls apart when the invariant feature in itself is fully predictive of the label.

**Constraint 2. (Identical invariant distribution.)** Across all $\mathcal{D} \in \mathbb{D}$, $p_\mathcal{D}(\mathbf{x}_{\text{inv}})$ is identical.
This constraint demands that the (marginal) invariant feature distribution must remain stable across domains (like in our binary-MNIST example). While this may appear to be unrealistic, (the exact distribution of the different types of cows and camels could vary across domains), we must emphasize that it is easier to make ERM fail when the invariant features are not stable (see example in App A Fig 4a).

**Constraint 3. (Conditional independence.)** For all $\mathcal{D} \in \mathbb{D}$, $\mathbf{x}_{\text{sp}} \perp\!\!\!\perp \mathbf{x}_{\text{inv}}|y$.
This constraint reflects the fact that in the MNIST example, we chose the class label and then picked the color feature independent of the actual hand-written digit picked from that class. This prevents us from designing complex relationships between the background and the object shape to show failure (see example in App A Fig 4b or the failure mode in Khani & Liang (2021)).

**Constraint 4. (Two-valued spurious features.)** We set $\mathcal{X}_{\text{sp}} = \mathbb{R}$ and the support of $x_{\text{sp}}$ in $\mathcal{D}_{\text{train}}$ is $\{-\mathcal{B}, +\mathcal{B}\}$.
This constraint[2] captures the simplicity of the cow-camel example where the background color is limited to yellow/green. Notably, this excludes failure borne out of high-dimensional (Tsipras et al., 2019) or carefully-constructed continuous-valued (Sagawa et al., 2020b; Shah et al., 2020) spurious features.

**Constraint 5. (Identity mapping.)** $\Phi$ is the identity mapping i.e., $\mathbf{x} = (\mathbf{x}_{\text{inv}}, \mathbf{x}_{\text{sp}})$.
This final constraint[3], also implicitly made in Sagawa et al. (2020b); Tsipras et al. (2019), prevents ERM from failing because of a hard-to-disentangle representation (see example in App A Fig 4c).

Before we connect these constraints to our main goal, it is worth mentioning their value beyond that goal. First, as briefly discussed above and elaborated in App A, each of these constraints in itself corresponds to a unique failure mode of ERM, one that is worth exploring in future work. Second, the resulting class of easy-to-learn tasks provides a theoretical (and a simplified empirical) test-bed that would help in broadly reasoning about OoD generalization. For example, any algorithm for solving OoD generalization should at the least hope to solve these easy-to-learn tasks well.

**Why is it hard to show that ERM relies on the spurious feature in easy-to-learn tasks?** Consider the simplest easy-to-learn 2D task. Specifically, during training we set $x_{\text{inv}} = y$ (and so Constraint 1 is satisfied) and $x_{\text{sp}}$ to be $y\mathcal{B}$ with probability $p \in [0.5, 1)$ and $-y\mathcal{B}$ with probability $1 - p$ (hence satisfying both Constraint 3 and 4). During test-time, the only shifts allowed are on the distribution of $x_{\text{sp}}$ (to respect Constraint 2). Observe from Fig 1b that this distribution has a support of the four points in $\{-1, +1\} \times \{-\mathcal{B}, +\mathcal{B}\}$ and is hence an abstract form of the cow-camel dataset which also has four groups of points: (a majority of) cows/camels against grass/sand and (a minority of) cows/camels against sand/grass. Fitting a max-margin classifier on $\mathcal{D}_{\text{train}}$ leads us to a simple yet key observation: *owing to the geometry of the four groups of points, the max-margin classifier has no reliance on the spurious feature despite the spurious correlation*. In other words, even though this dataset distills what seem to be the core aspects of the cow/camel dataset, we are unable to reproduce the corresponding behavior of ERM. In the next two sections, we will try to resolve this apparent paradox.

---

[2]Note that the discrete-value restriction holds only during training. This is so that, in our experiments, we can study two kinds of test-time shifts, one within the support $\{-\mathcal{B}, +\mathcal{B}\}$ and one outside of it (the vulnerability to both of which boils down to the level of reliance on $x_{\text{sp}}$).

[3]The rest of our discussion would hold even if $\Phi$ corresponds to any orthogonal transformation of $(\mathbf{x}_{\text{inv}}, \mathbf{x}_{\text{sp}})$, since the algorithms we study are rotation-invariant. But for simplicity, we'll work with the identity mapping.

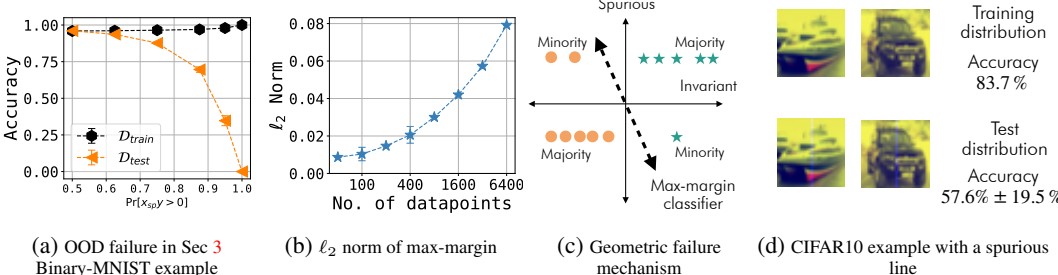

(a) OOD failure in Sec 3 Binary-MNIST example

(b) $\ell_2$ norm of max-margin

(c) Geometric failure mechanism

(d) CIFAR10 example with a spurious line

Figure 2: **Geometric skews: Fig 2a** demonstrates failure of the max-margin classifier in the easy-to-learn Binary-MNIST task of Sec 3. **Fig 2b** forms the basis for our theory in Sec 4 by showing that the random-features-based max-margin has norms increasing with the size of the dataset. In **Fig 2c**, we visualize the failure mechanism arising from the "geometric skew": with respect to the $\mathbf{x}_{\text{inv}}$-based classifier, the closest minority point is farther away than the closest majority point, which crucially tilts the max-margin classifier towards $w_{\text{sp}} > 0$. **Fig 2d**, as further discussed in Sec 4, stands as an example for how a wide range of OoD failures can be explained geometrically.

## 4 FAILURE DUE TO GEOMETRIC SKEWS

A pivotal piece in solving this puzzle is a particular geometry underlying how the invariant features in real-world distributions are separated. To describe this, consider the random features representation of MNIST (i.e., $\mathbf{x}_{\text{inv}}$) and fit a max-margin classifier on it, i.e., the least-norm $\mathbf{w}_{\text{inv}}$ that achieves a margin of $y \cdot (\mathbf{w}_{\text{inv}} \cdot \mathbf{x}_{\text{inv}} + b) \geq 1$ on all data (for some $b$). Then, we'd observe that *as the number of training points increase, the $\ell_2$ norm of this max-margin classifier grows* (see Figure 2b); similar observations also hold for CIFAR10 (see App C.2). This observation (which stems from the geometry of the dataset) builds on ones that were originally made in Neyshabur et al. (2017); Nagarajan & Kolter (2017; 2019) for norms of overparameterized neural networks. Our contribution here is to empirically establish this for a linear overparameterized model and then to theoretically relate this to OoD failure. In the following discussion, we will take this "increasing norms" observation as a given[4]), and use it to explain why the max-margin classifier trained on all the features (including $x_{\text{sp}}$) relies on the spurious feature.

Imagine every input $\mathbf{x}_{\text{inv}}$ to be concatenated with a feature $x_{\text{sp}} \in \{-\mathcal{B}, \mathcal{B}\}$ that is spuriously correlated with the label i.e., $\Pr[x_{\text{sp}} \cdot y > 0] > 0.5$. The underlying spurious correlation implicitly induces two disjoint groups in the dataset $S$: a *majority group* $S_{\text{maj}}$ where $x_{\text{sp}} \cdot y > 0$ e.g., cows/camels with green/yellow backgrounds, and a *minority group* $S_{\text{min}}$ where $x_{\text{sp}} \cdot y < 0$ e.g., cows/camels with yellow/green backgrounds. Next, let $\mathbf{w}_{\text{all}} \in \mathcal{X}_{\text{inv}}$ denote the least-norm vector that (a) lies in the invariant space and (b) classifies all of $S$ by a margin of at least 1. Similarly, let $\mathbf{w}_{\text{min}} \in \mathcal{X}_{\text{inv}}$ denote a least-norm, purely-invariant vector that classifies all of $S_{\text{min}}$ by a margin of at least 1. Crucially, since $S_{\text{min}}$ has much fewer points than $S$, by the "increasing-norm" property, we can say that $\|\mathbf{w}_{\text{min}}\| \ll \|\mathbf{w}_{\text{all}}\|$. We informally refer to the gap in these $\ell_2$ norms as a geometric skew.

Given this skew, we explain why the max-margin classifier must use the spurious feature. One way to classify the data is to use only the invariant feature, which would cost an $\ell_2$ norm of $\|\mathbf{w}_{\text{all}}\|$. But there is another alternative: use the spurious feature as a short-cut to classify the majority of the dataset $S_{\text{maj}}$ (by setting $w_{\text{sp}} > 0$) and combine it with $\mathbf{w}_{\text{min}}$ to classify the remaining minority $S_{\text{min}}$. Since $\|\mathbf{w}_{\text{min}}\| \ll \|\mathbf{w}_{\text{all}}\|$, the latter strategy requires lesser $\ell_2$ norm, and is therefore the strategy opted by the max-margin classifier. We illustrate this failure mechanism in a 2D dataset in Fig 2c. Here, we have explicitly designed the data to capture the "increasing norms" property: the distance between purely-invariant classifier boundary (i.e., a vertical separator through the origin) and the closest point in $S_{\text{maj}}$ is smaller than that of the closest point in $S_{\text{min}}$. In other words, a purely-invariant classifier

---

[4]To clarify, we take the "increasing norms" observation as a given in that we don't provide an explanation for why it holds. Intuitively, we suspect that norms increase because as we see more datapoints, we also sample rarer/harder training datapoints. We hope future work can understand this explain better.

would require much greater norm to classify the majority group by a margin of 1 than to classify the minority group. We can then visually see that the max-margin classifier would take the orientation of a diagonal separator that uses the spurious feature rather than a vertical, purely-invariant one.

The following result formalizes the above failure. In particular, for any arbitrary easy-to-learn task, we provide lower and upper bounds on $w_{sp}$ that are larger for smaller values of $\|\mathbf{w}_{min}\|/\|\mathbf{w}_{all}\|$. For readability, we state only an informal, version of our theorem below. In App B.1, we present the full, precise result along with the proof.

**Theorem 1.** *(informal) Let* $\mathbb{H}$ *be the set of linear classifiers,* $h(x) = \mathbf{w}_{inv}\mathbf{x}_{inv} + w_{sp}x_{sp} + b$. *Then for any task satisfying all the constraints in Sec 3.1 with* $\mathcal{B} = 1$, *the max-margin classifier satisfies:*

$$1 - 2\sqrt{\|\mathbf{w}_{min}\|/\|\mathbf{w}_{all}\|} \leq w_{sp} \leq \frac{1}{\|\mathbf{w}_{min}\|/\|\mathbf{w}_{all}\|} - 1.$$

A salient aspect of this result is that it explains the varying dynamics between the underlying spurious correlation and spurious-feature-reliance in the classifier. First, as the correlation increases ($Pr_{\mathcal{D}_{train}}[x_{sp} \cdot y > 0] \to 1.0$), the size of the minority group decreases to zero. Then, we empirically know that $\|\mathbf{w}_{min}\|/\|\mathbf{w}_{all}\|$ progressively shrinks all the way down to 0. Then, we can invoke the lower bound which implies that $w_{sp}$ grows to $\approx 1$. This implies serious vulnerability to the test-time shifts: any flip in the sign of the spurious feature can reduce the original margin of $\approx 1$ by a value of $2|w_{sp}x_{sp}| \approx 2$ (since $\mathcal{B} = 1$ here) making the margin negative (implying misclassification). On the other hand, when spurious correlations diminish ($Pr_{\mathcal{D}_{train}}[x_{sp} \cdot y > 0] \to 0.5$), the value of $\|\mathbf{w}_{min}\|$ grows comparable to $\|\mathbf{w}_{all}\|$, and our upper bound suggests that the spurious component must shrink towards $\approx 0$, thereby implying robustness to these shifts.

**Broader empirical implications.** While our theorem explains failure in linear, easy-to-learn settings, the underlying geometric argument can be used to intuitively understand failure in more general settings i.e., setting where the classifier is non-linear and/or the task is not easy-to-learn. For illustration, we identify a few such unique non-linear tasks involving the failure of a neural network. The first two tasks below can be informally thought of as easy-to-learn tasks[5]:

- In Fig 1c, we consider a CIFAR10 task where we add a line to with its color spuriously correlated with the class only during training. The $\gtrsim 10\%$ OoD accuracy drop of a ResNet here, we argue (in App C.3.1), arises from the fact that it takes greater norms for the ResNet to fit larger proportions CIFAR10.
- In Fig 1d, we consider a colored Cats vs. Dogs task (Elson et al., 2007), where a majority of the datapoints are blue-ish and a minority are green-ish. During testing, we color all datapoints to be green-ish. Crucially, even though there is no correlation between the label and the color of the images, the OoD accuracy of an ResNet drops by $\gtrsim 20\%$. To explain this, in App. C.3.3, we identify an "implicit", non-visual kind of spurious correlation in this dataset, one between the label and a particular component of the difference between the two channels.

Next, we enumerate two not-easy-to-learn tasks. Here, one of the easy-to-learn constraints is disobeyed significantly enough to make the task hard, and this is essential in causing failure. In other words, the failure modes here correspond to ones that were outlined in Sec 3.1. Nevertheless, we argue in App C.3 that even these modes can be reasoned geometrically as a special case of Theorem 1. The two not-easy-to-learn tasks are as follows:

- In Fig 2d, we add a line to the last channel of CIFAR10 images regardless of the label, and make the line brighter during testing resembling a camera glitch, which results in a $\gtrsim 27\%$ drop in a ResNet's accuracy. We geometrically argue how this failure arises from breaking Constraint 5.
- In App C.3.5, we consider an MNIST setting inspired by Tsipras et al. (2019), where failure arises (geometrically) due to high-dimensional spurious features (breaking Constraint 4).

We hope that these examples, described in greater detail in App C.3, provide (a) a broader way to think about how spurious correlations manifest, and (b) how a variety of resulting failure modes can be reasoned geometrically.

---

[5]Note that although technically speaking these tasks do break Constraint 4 (as the spurious feature does not take two discrete values) this is not essential to the failure.

## 5 FAILURE DUE TO STATISTICAL SKEWS

Having theoretically studied max-margin classifiers, let us now turn our attention to studying linear classifiers trained by gradient descent on logistic/exponential loss. Under some conditions, on linearly separable datasets, these classifiers would converge to the max-margin classifier given infinite time (Soudry et al., 2018; Ji & Telgarsky, 2018). So it is reasonable to say that even these classifiers would suffer from the geometric skews, even if stopped in some finite time. However, are there any other failure modes that would arise here?

To answer this, let us dial back to the easiest-to-learn task: the setting with four points $\{-1, +1\} \times \{-\mathcal{B}, +\mathcal{B}\}$, where in the training distribution (say $\mathcal{D}_{\text{2-dim}}$), we have $x_{\text{inv}} = y$ and $x_{\text{sp}}$ to be $y\mathcal{B}$ with probability $p \in [0.5, 1)$ and $-y\mathcal{B}$ with probability $1-p$. Here, even though the max-margin does not rely on $x_{\text{sp}}$ for any level of spurious correlation $p \in [0.5, 1)$ — there are no geometric skews here after all — the story is more complicated when we empirically evaluate via gradient descent stopped in finite time $t$. Specifically, for various values of $p$ we plot $w_{\text{sp}}/\sqrt{w_{\text{inv}}^2 + w_{\text{sp}}^2}$ vs. $t$ (here, looking at $w_{\text{sp}}$ alone does not make sense since the weight norm grows unbounded). We observe in Fig 3a that the spurious component appears to stagnate around a value proportional to $p$, even after sufficiently long training, and even though it is supposed to converge to $0$. Thus, even though max-margin doesn't fail in this dataset, finite-time-stopped gradient descent fails. Why does this happen?

To explain this behavior, a partial clue already exists in Soudry et al. (2018); Ji & Telgarsky (2018): gradient descent can have a frustratingly slow logarithmic rate of convergence to the max-margin i.e,. the ratio $|w_{\text{sp}}/w_{\text{inv}}|$ could decay to zero as slow as $1/\ln t$. However, this bound is a distribution-independent one that does not explain why the convergence varies with the spurious correlation. To this end, we build on this result to derive a distribution-specific convergence bound in terms of $p$, that applies to any easy-to-learn task (where $\mathbf{x}_{\text{inv}}$ may be higher dimensional unlike in $\mathcal{D}_{\text{2-dim}}$). For convenience, we focus on continuous-time gradient descent under the exponential loss $\exp(-yh(\mathbf{x}))$ (the dynamics of which is similar to that of logistic loss as noted in Soudry et al. (2018)). Then we consider any easy-to-learn task and informally speaking, any corresponding dataset without geometric skews, so the max-margin wouldn't rely on the spurious feature. We then study the convergence rate of $w_{\text{sp}}(t) \cdot \mathcal{B}/\mathbf{w}_{\text{inv}}(t) \cdot \mathbf{x}_{\text{inv}}$ to $0$ i.e., the rate at which the ratio between the output of the spurious component to that of the invariant component converges to its corresponding max-margin value. We show that the convergence rate is $\Theta(1/\ln t)$, crucially scaled by an extra factor that monotonically increases in $[0, \infty)$ as a function of the spurious correlation, $p \in [0.5, 1)$, thus capturing slower convergence for larger spurious correlation. Another notable aspect of our result is that when there is no spurious correlation ($p = 0.5$), both the upper and lower bound reduce to $0$, indicating quick convergence. We provide the full statement and proof of this bound in App B.2. For completeness, we also provide a more precise analysis of the dynamics for a 2D setting under both exponential and logistic loss in Theorem 5 and Theorem 6 in App B.2.

**Theorem 2.** *(informal) Let $\mathbb{H}$ be the set of linear classifiers $h(\mathbf{x}) = \mathbf{w}_{\text{inv}} \cdot \mathbf{x}_{\text{inv}} + w_{\text{sp}}x_{\text{sp}}$. Then, for any easy-to-learn task, and for any dataset without geometric skews, continuous-time gradient descent training of $\mathbf{w}_{\text{inv}}(t) \cdot \mathbf{x}_{\text{inv}} + w_{\text{sp}}(t)x_{\text{sp}}$ to minimize the exponential loss, satisfies:*

$$\Omega\left(\frac{\ln\frac{1+p}{1+\sqrt{p(1-p)}}}{\ln t}\right) \leq \frac{w_{\text{sp}}(t)\mathcal{B}}{|\mathbf{w}_{\text{inv}}(t) \cdot \mathbf{x}_{\text{inv}}|} \leq \mathcal{O}\left(\frac{\ln\frac{p}{1-p}}{\ln t}\right), \quad \text{where } p := Pr_{\mathcal{D}_{\text{train}}}[x_{\text{sp}} \cdot y > 0] \in [0.5, 1).$$

The intuition behind this failure mode is that in the initial epochs, when the loss $\exp(-y\mathbf{w} \cdot \mathbf{x})$ on all points are more or less the same, the updates $1/|S| \cdot \sum_{(\mathbf{x},y)\in S} y\mathbf{x}\exp(-y\mathbf{w}\cdot\mathbf{x})$ roughly push along the direction, $1/|S| \cdot \sum_{(\mathbf{x},y)\in S} y\mathbf{x}$. This is the precise (mis)step where gradient descent "absorbs" the spurious correlation, as this step pushes $w_{\text{sp}}$ along $p\mathcal{B} - (1-p)\mathcal{B} = (2p-1)\mathcal{B}$. While this update would be near-zero when there is only little spurious correlation ($p \approx 0.5$), it takes larger values for larger levels of spurious correlation ($p \approx 1$). Unfortunately, under exponential-type losses, the gradients decay with time, and so the future gradients, even if they eventually get rid of this absorbed spurious component, take an exponentially long time to do so.

**Broader empirical implications.** We now demonstrate the effect of statistical skews in more general empirical settings consisting of a non-linear easy-to-learn task learned using a neural network.

---

[6]The overall accuracy on CIFAR10 is low because even though $|S_{\text{con}}| = 50k$ and $|S_{\text{exp}}| = 455k$, the number of unique samples here is just $5k$. See App C.4.2 for more explanation.

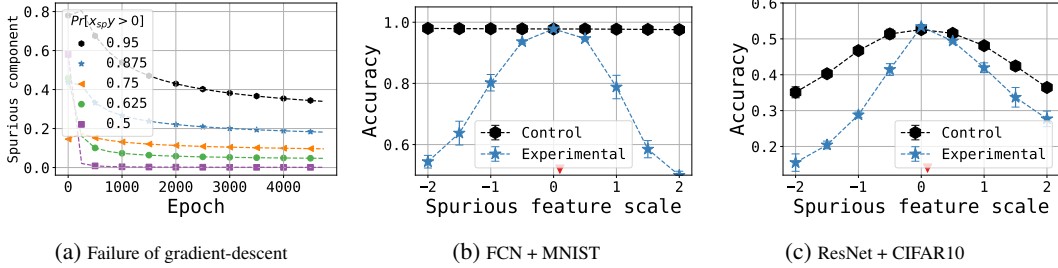

(a) Failure of gradient-descent     (b) FCN + MNIST     (c) ResNet + CIFAR10

Figure 3: **Statistical skews:** In **Fig 3a**, we plot the slow convergence of $w_{\text{sp}}/\sqrt{w_{\text{inv}}^2 + w_{\text{sp}}^2} \in [0, 1]$ under logistic loss with learning rate $0.001$ and a training set of $2048$ from $\mathcal{D}_{\text{2-dim}}$ with $\mathcal{B} = 1$. In **Fig 3b** and **Fig 3c**, we demonstrate the effect of statistical skews in neural networks. Here, during test-time we shift both the scale of the spurious feature (from its original scale of $0.1$ as marked by the red triangle) and its correlation. Observe that the network trained on the statistically-skewed $S_{\text{exp}}$ is more vulnerable to these shifts compared to no-skew $S_{\text{con}}$.[6]

Isolating the statistical skew effect is however challenging in practice: any gradient-descent-trained model is likely to be hurt by both statistical skews and geometric skews, and we'd have to somehow disentangle the two effects. We handle this by designing the following experiment. We first create a control dataset $S_{\text{con}}$ where there are no geometric or statistical skews. For this, we take a set of images $S_{\text{inv}}$ (with no spurious features), and create two copies of it, $S_{\text{maj}}$ and $S_{\text{min}}$ where we add spurious features, positively and negatively aligned with the label, respectively, and define $S_{\text{con}} = S_{\text{maj}} \cup S_{\text{min}}$. Next, we create an experimental dataset $S_{\text{exp}}$ with a statistical skew in it. We do this by taking $S_{\text{con}}$ and duplicating $S_{\text{maj}}$ in it so that the ratio $|S_{\text{maj}}| : |S_{\text{min}}|$ becomes $10 : 1$. Importantly, this dataset has no geometric skews, since merely replicating points does not affect the geometry. Then, if we were to observe that (stochastic) gradient descent on $S_{\text{exp}}$ results in greater spurious-feature-reliance than $S_{\text{con}}$, we would have isolated the effect of statistical skews.

Indeed, we demonstrate this in two easy-to-learn tasks. First, we consider a Binary-MNIST task learned by an fully-connected network. Here, we concatenate a spurious channel where either all the pixels are "on" or "off". Second, we consider a (multiclass) CIFAR10 task (where we add a spuriously colored line) learned using a ResNet (He et al., 2016). In Fig 3, we demonstrate that training on $S_{\text{exp}}$ leads to less robust models than on $S_{\text{con}}$ in both these tasks. In other words, gradient descent on $S_{\text{exp}}$ leads to greater spurious-feature-reliance, thus validating the effect of statistical skews in practice. More details are provided in App C.4.

## 6 CONCLUSIONS AND FUTURE WORK

We identify that spurious correlations during training can induce two distinct skews in the training set, one geometric and another statistical. These skews result in two complementary ways by which empirical risk minimization (ERM) via gradient descent is guaranteed to rely on those spurious correlations. At the same time, our theoretical results (in particular, the upper bounds on the spurious component of the classifier) show that when these skews do disappear, there is no failure within the considered tasks. This suggests that within the class of easy-to-learn tasks and for gradient-descent-trained linear models, the above discussion likely captures all possible failure modes.

However, when we do venture into the real-world to face more complicated tasks and use non-linear, deep models, many other kinds of failure modes would crop up (such as the ones we enumerate in Sec 3.1, in addition to the fundamental ones mentioned above). Indeed, the central message of our work is that there is no one unique mechanism by which classifiers fail under spurious correlations, even in the simplest of tasks. This in turn has a key practical implication: in order to improve our solutions to OoD generalization, it would be valuable to figure out whether or not a unified solution approach is sufficient to tackle all these failure mechanisms. While we outline some solutions in App D, we hope that the foundation we have laid in this study helps future work in better tackling out-of-distribution challenges.

**Acknowledgements.** We thank Hanie Sedghi for providing useful feedback on the draft.

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

## A  CONSTRAINTS AND OTHER FAILURE MODES

Here, we elaborate on how each of the constraints we impose on the easy-to-learn tasks corresponds to eliminating a particular complicated kind of failure of ERM from happening. In particular, for each of these constraints, we'll construct a task that betrays that constraint (but obeys all the others) and that causes a unique kind of failure of ERM. We hope that laying these out concretely can provide a useful starting point for future work to investigate these various failure modes. Finally, it is worth noting that all the failure modes here can be explained via a geometric argument, and furthermore the failure modes for Constraint 5 and Constraint 4 can be explained as a special case of the argument in Theorem 1.

**Failure due to weakly predictive invariant feature (breaking Constraint 1).** We enforced in Constraint 1 that the invariant feature be fully informative of the label. For a setting that breaks this

constraint, consider a 2D task with *noisy* invariant features, where across all domains, we have a Gaussian invariant feature of the form $x_{\text{inv}} \sim \mathcal{N}(y, \sigma^2_{\text{inv}})$ —- this sort of a noisy invariant feature was critically used in Tsipras et al. (2019); Sagawa et al. (2020b); Arjovsky et al. (2019) to explain failure of ERM. Now, assume that during training we have a spurious feature $x_{\text{sp}} \sim \mathcal{N}(y, \sigma^2_{\text{sp}})$ (say with relatively larger variance, while positively correlated with $y$). Then, observe that the Bayes optimal classifier on $\mathcal{D}_{\text{train}}$ is sgn $(x_{\text{inv}}/\sigma_{\text{inv}} + x_{\text{sp}}/\sigma_{\text{sp}})$ i.e., it must rely on the spurious feature.

However, also observe that if one were to eliminate noise in the invariant feature by setting $\sigma_{\text{inv}} \to 0$ (thus making the invariant feature *perfectly informative* of the label like in our MNIST example), the Bayes optimal classifier approaches sgn $(x_{\text{inv}})$, thus succeeding after all.

**Failure due to "unstable" invariant feature (breaking Constraint 2).** If during test-time, we push the invariant features closer to the decision boundary (e.g., partially occlude the shape of every camel and cow), test accuracy will naturally deteriorate, and this embodies a unique failure mode.

Concretely, consider a domain generalization task where across all domains $x_{\text{inv}} \geq -0.5$ determines the true boundary (see Fig 4a). Now, assume that in the training domains, we see $x_{\text{inv}} = 2y$ or $x_{\text{inv}} = 3y$. This would result in learning a max-margin classifier of the form $x_{\text{inv}} \geq 0$. Now, during test-time, if one were to provide "harder" examples that are closer to the true boundary in that $x_{\text{inv}} = -0.5 + 0.1y$, then all the positive examples would end up being misclassified.

**Failure due to complex conditional dependencies (breaking Constraint 3).** This constraint imposed $x_{\text{inv}} \perp\!\!\!\perp x_{\text{sp}}|y$. Stated more intuitively, given that a particular image is that of a camel, this constraint captures the fact that knowing the background color does not tell us too much about the precise shape of the camel. An example where this constraint is broken is one where $x_{\text{inv}}$ and $x_{\text{sp}}$ share a neat geometric relationship during training, but not during testing, which then results in failure as illustrated below.

Consider the example in Fig 4b where for the positive class we set $x_{\text{inv}} + x_{\text{sp}} = 1$ and for the negative class we set $x_{\text{inv}} + x_{\text{sp}} = -1$, thereby breaking this constraint. Now even though the invariant feature in itself is fully informative of the label, while the spurious feature is not, the max-margin here is parallel to the line $x_{\text{inv}} + x_{\text{sp}} = c$ and is therefore reliant on the spurious feature.

**Failure mode due to high-dimensional spurious features (lack of Constraint 4).** Akin to the setting in Tsipras et al. (2019), albeit without noise, consider a task where the spurious feature has $D$ different co-ordinates, $\mathcal{X}_{\text{sp}} = \{-1, 1\}^D$ and the invariant feature just one $\mathcal{X}_{\text{inv}} = \{-1, +1\}$. Then, assume that the $i$th spurious feature $x_{\text{sp},i}$ independently the value $y$ with probability $p_i$ and $-y$ with probability $1 - p_i$, where without loss of generality $p_i > 1/2$. Here, with high probability, all datapoints in $S$ can be separated simply by summing up the spurious features (given $D$ is large enough). Then, we argue that the max-margin classifier would provide some non-zero weight to this direction because it helps maximize its margin (see visualization in Fig 4d).

One way to see why this is true is by invoking a special case of Theorem 1. In particular, if we define $\sum x_{\text{sp},i}$ to be a single dimensional spurious feature $x_{\text{sp}}$, this feature satisfies $x_{\text{sp}} \cdot y > 0$ on all training points. In other words, this is an extreme scenario with no minority group. Then, Theorem 1 would yield a positive lower bound on the weight given to $x_{\text{sp}}$, explaining why the classifier relies on the spurious pixels. For the sake of completeness, we formalize all this discussion below:

**Proposition 1.** *Let $c$ be a constant such that for all $i$, $p_j > \frac{1}{2} + \frac{c}{2}$. Let $D$ be sufficiently large so that $D \geq \frac{1}{2c}\sqrt{2 \ln \frac{m}{\delta}}$ where $m$ is the number of training datapoints in $S$. Then, w.h.p. of $1 - \delta$ over the draws of $S$, the max-margin classifier corresponds is of the form $w_{\text{inv}} x_{\text{inv}} + \mathbf{w}_{\text{sp}} \mathbf{x}_{\text{sp}}$ where:*

$$\frac{\|\mathbf{w}_{\text{sp}}\|}{w_{\text{inv}}} \geq \frac{c\sqrt{D}}{2}.$$

*Proof.* First, we'll show that on $S$ there exists a classifier that relies purely on the spurious features to separate the data. In particular, consider $\mathbf{w}'_{\text{sp}}$ where the $i$th dimension is $1/\sqrt{D}$ if $p_i > 1/2$, and $-1/\sqrt{D}$ otherwise. By the Hoeffding's inequality, we have that with high probability $1 - \delta$, on all the $m$ training datapoints, $y\mathbf{w}'_{\text{sp}} \cdot \mathbf{x}_{\text{sp}} \geq \frac{1}{\sqrt{D}} \sum(2p_i - 1) - \sqrt{\frac{2}{D} \ln \frac{m}{\delta}} \geq \frac{c}{2}\sqrt{D}$.

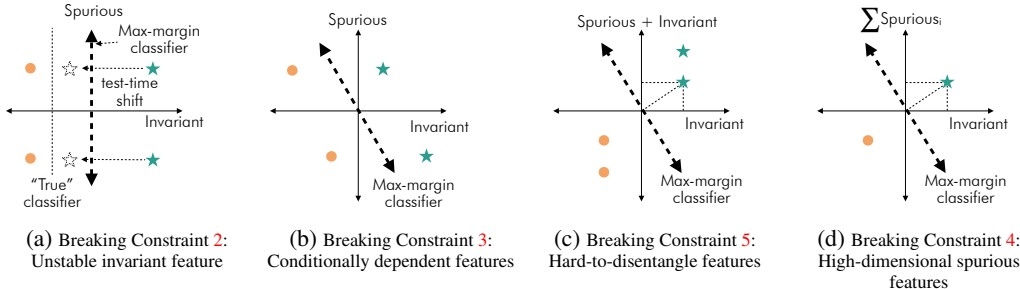

(a) Breaking Constraint 2: Unstable invariant feature

(b) Breaking Constraint 3: Conditionally dependent features

(c) Breaking Constraint 5: Hard-to-disentangle features

(d) Breaking Constraint 4: High-dimensional spurious features

Figure 4: **Other failure modes:** We visualize the different (straightforward) ways in which a max-margin classifier can be shown to fail in tasks where one of the Constraints in Sec 3.1 is disobeyed. More discussion in Sec A.

Now, for the max-margin classifier, assume that $w_{\text{inv}}^2 = \alpha$. Further, assume that the margin contributed by $\mathbf{w}_{\text{sp}} x_{\text{sp}} + b$ equals $\sqrt{1-\alpha^2} m$ for some $m$. Observe that $m$ must satisfy $m \geq \frac{c}{2}\sqrt{D}$ (as otherwise, we can replace $\mathbf{w}_{\text{sp}}$ with $\sqrt{1-\alpha^2}\mathbf{w}'_{\text{sp}}$ to achieve a better margin). Now, for the resulting margin to be maximized, $\alpha$ must satisfy $\frac{\alpha}{\sqrt{1-\alpha^2}} = \frac{1}{m}$.

$\square$

**Failure mode due to hard-to-separate features (lack of Constraint 5)** Here we assumed that the feature space can be orthogonally decomposed into invariant and spurious features. Now let us imagine a 2D task, visualized in Fig 4c where this is not respected in that each datapoint is written as $(x_{\text{inv}}, x_{\text{inv}} + x_{\text{sp}})$, assuming that $x_{\text{inv}} = y$ and $x_{\text{sp}} \in \{-0.5, 0.5\}$. A practical example of this sort of structure is the example in Fig 2d where we add a line to the last channel of CIFAR10, and then vary the brightness of the line during test-time.

To understand why failure occurs in this 2D example, observe that, regardless of the correlation between $x_{\text{sp}}$ and $y$, we'd have that $(x_{\text{inv}} + x_{\text{sp}}) \cdot y > 0$. In other words, the second co-ordinate is fully informative of the label. The max-margin classifier, due to its bias, relies on both the first and the second co-ordinate to maximize its margin i.e., the classifier would be of the form $w_1 x_{\text{inv}} + w_2(x_{\text{inv}} + x_{\text{sp}})$ where $w_2 > 0$. (Again, like in our discussion of the failure mode of Constraint 4, we can argue this via Thm 1 by considering $x_{\text{inv}} + x_{\text{sp}}$ itself as a spurious feature, and by observing that there's no minority group here.) Hence, by assigning a positive weight to the second co-ordinate it inadvertently becomes susceptible to the spurious feature that may shift during testing.

# B  PROOFS

## B.1  PROOF OF THEOREM 1 ON FAILURE DUE GEOMETRIC SKEWS

Below we provide a proof for our result analyzing the failure mode arising from geometric skews in the data.

Recall that given a dataset $S$, where the spurious feature can take only values in $\{-\mathcal{B}, +\mathcal{B}\}$, we partitioned $S$ into two subsets $S_{\text{maj}}$ and $S_{\text{min}}$ where in $S_{\text{maj}}$ the points satisfy $x_{\text{sp}} \cdot y > 0$ and in $S_{\text{min}}$ the points satisfy $x_{\text{sp}} \cdot y < 0$.

Next, we define two key notations. First, for any dataset $T \subseteq S$, let $\mathbf{v}(T) \in \mathcal{X}_{\text{inv}}$ denote a least-norm vector (purely in the invariant space) that achieves a margin of at least 1 on all datapoints in $T$ (we'll define this formally a few paragraphs below). Similarly, let $\tilde{\mathbf{v}}(T) \in \mathcal{X}_{\text{inv}}$ denote a least-norm vector that achieves a margin of at least 1 on $T$, and a margin of at least 0 on $S \setminus T$ (again, full definition will shortly follow). While by definition, $\|\mathbf{v}(T)\| \leq \|\tilde{\mathbf{v}}(T)\|$, we can informally treat these quantities as the same since empirically $\|\mathbf{v}(T)\| \approx \|\tilde{\mathbf{v}}(T)\|$. We show these plots in Sec C.1 for both MNIST and CIFAR10. But importantly, both these quantities grow with the size of $T$. Then, by virtue of the small size of the minority group $S_{\text{min}}$, we can say that $\|\mathbf{v}(S_{\text{min}})\|$ is smaller than both $\|\mathbf{v}(S_{\text{maj}})\|$ and $\|\mathbf{v}(S)\|$. We refer to this gap as a geometric skew. When this skew is prominent enough (e.g.,

$\|\mathbf{v}(S_{\min})\|/\|\mathbf{v}(S)\| \approx 0$), our result below argues that the spurious component in the overall max-margin classifier must be sufficiently large (and positive). On the flip side, we also show that when the skew is negligible enough (e.g., $\|\mathbf{v}(S_{\min})\|/\|\mathbf{v}(S)\| \approx 1$), then the spurious component has to be sufficiently small.

To be able to better visualize these bounds, we write these as bounds on $|\mathcal{B}w_{\text{sp}}|$ (i.e., $|w_{\text{sp}}x_{\text{sp}}|$ rather than $|w_{\text{sp}}|$). Then we can think of a lower bound of the form $|\mathcal{B}w_{\text{sp}}| \gtrsim 1$ as demonstrating serious failure as a shift in the correlation can adversely reduce the original margin of $\approx 1$.

Before we state the result, for clarity, we state the full mathematical definition of $\mathbf{v}$ and $\tilde{\mathbf{v}}$ as follows. For any $T \subseteq S$:

$$\mathbf{v}(T), b(T) = \underset{\mathbf{w}_{\text{inv}} \in \mathcal{X}_{\text{inv}}, b}{\arg\min} \quad \|\mathbf{w}_{\text{inv}}\|^2$$
$$\text{s.t.} \quad y(\mathbf{w}_{\text{inv}} \cdot \mathbf{x}_{\text{inv}}) + b) \geq 1 \qquad \forall\left((\mathbf{x}_{\text{inv}}, x_{\text{sp}}), y\right) \in T$$

$$\tilde{\mathbf{v}}(T), \tilde{b}(T) = \underset{\mathbf{w}_{\text{inv}} \in \mathcal{X}_{\text{inv}}, b}{\arg\min} \quad \|\mathbf{w}_{\text{inv}}\|^2$$
$$\text{s.t.} \quad y(\mathbf{w}_{\text{inv}} \cdot \mathbf{x}_{\text{inv}}) + b) \geq 1 \qquad \forall\left((\mathbf{x}_{\text{inv}}, x_{\text{sp}}), y\right) \in T$$
$$\qquad\quad y(\mathbf{w}_{\text{inv}} \cdot \mathbf{x}_{\text{inv}}) + b) \geq 0 \qquad \forall\left((\mathbf{x}_{\text{inv}}, x_{\text{sp}}), y\right) \in S \setminus T$$

Using these notations, we state our full theorem and provide its proof below.

**Theorem 3.** *Let $\mathbb{H}$ be the set of linear classifiers, $h(x) = \mathbf{w}_{\text{inv}}\mathbf{x}_{\text{inv}} + w_{\text{sp}}x_{\text{sp}} + b$. Let the geometric skews in a dataset $S$ be quantified through the terms $\kappa_1 = \|\mathbf{v}(S_{\min})\|/\|\mathbf{v}(S)\|$, $\kappa_2 = \|\mathbf{v}(S_{\min})\|/\|\mathbf{v}(S_{\text{maj}})\|$ and $\tilde{\kappa}_1 := \|\tilde{\mathbf{v}}(S_{\min})\|/\|\tilde{\mathbf{v}}(S)\|$, $\tilde{\kappa}_2 := \|\tilde{\mathbf{v}}(S_{\min})\|/\|\tilde{\mathbf{v}}(S_{\text{maj}})\|$. Then for any task satisfying all the constraints in Sec 3.1 the max-margin classifier satisfies the inequalities (where for readability, we will use $c_1 := 1/(2\|\tilde{\mathbf{v}}(S)\|\mathcal{B})$, $c_2 := 1/(2\|\tilde{\mathbf{v}}(S_{\text{maj}})\|\mathcal{B})$):*

$$\mathcal{B}w_{\text{sp}} \geq \max\left(1 - 2\sqrt{\tilde{\kappa}_1 + c_1^2}, 0\right) \qquad\qquad \textit{if} \quad \tilde{\kappa}_2 \leq \sqrt{1/4 - c_2^2}, \quad \textit{and}$$
$$|\mathcal{B}w_{\text{sp}}| \leq \min\left(1/\kappa_1 - 1, \mathcal{B}\|\mathbf{v}(S)\|\right) \qquad\qquad \textit{if} \quad \kappa_2 \leq 1.$$

For readability, it helps to think of $c_1$ and $c_2$ as small constants here (also see remark below). Furthermore, for readability, one can also imagine that all the $\kappa$ terms are numerically similar to each other.

We make a few remarks below before providing the proof.

**Remark 1.** For the lower bound on $w_{\text{sp}}$ to be positive, we need $c_1$ and $c_2$ to be small. This would be true when either $\mathcal{B}$ or $\|\mathbf{v}(S)\|$ (or $\|\tilde{\mathbf{v}}(S_{\text{maj}})\|$ ) is sufficiently large. This is intuitive: after all, if $\mathcal{B}$ is too small (say 0) there is no spurious feature effectively and therefore the max-margin has no incentive to use it; similarly, if $\mathbf{v}(S)$ is too small (say 0), then the max-margin has no incentive to use any feature besides the invariant feature, which is already quite cheap.

**Remark 2.** The above result is not intended to be a numerically tight/upper lower bound. In fact, the proof can be tightened in numerous places which we however avoid, to keep the result and the proof simple. The bound is rather meant to be instructive of the effect of the geometric skew (i.e., the gap between the max-margin norms on the minority and whole/majority dataset) on the spurious component.

*Proof.* We present the proof of lower bound first, followed by the upper bound.

**Proof of lower bound.** First, we will show that there exists a classifier of norm 1 that relies on the spurious feature to create a sufficiently large margin. We'll let this classifier be of the form $\alpha \frac{\tilde{\mathbf{v}}(S_{\min})}{\|\tilde{\mathbf{v}}(S_{\min})\|} \cdot \mathbf{x}_{\text{inv}} + \sqrt{1 - \alpha^2}x_{\text{sp}} + \alpha b_{\min}$. By the definition of $\tilde{\mathbf{v}}(S_{\min})$, the margin of this classifier on any datapoint in $S_{\min}$ is at least $\frac{\alpha}{\|\tilde{\mathbf{v}}(S_{\min})\|} - \sqrt{1 - \alpha^2}\mathcal{B}$. Again, by the definition of $\tilde{\mathbf{v}}(S_{\min})$, the margin on $S_{\text{maj}}$ is at least $\sqrt{1 - \alpha^2}\mathcal{B}$. Let us pick an $\alpha$ such that these two quantities are equal. Such

an $\alpha$ would satisfy $\frac{\alpha}{\sqrt{1-\alpha^2}} = 2\|\tilde{\mathbf{v}}(S_{\min})\|\mathcal{B}$. By plugging this back, we get that the resulting margin of this classifier on the whole dataset $S$ is at least $\frac{\mathcal{B}}{\sqrt{1+4\|\tilde{\mathbf{v}}(S_{\min})\|^2\mathcal{B}^2}}$. In other words, this also means the least norm classifier $\mathbf{w}$ with a margin of at least $1$ on $S$ has its norm upper bounded as:

$$\|\mathbf{w}\| \leq \frac{\sqrt{1+4\|\tilde{\mathbf{v}}(S_{\min})\|^2\mathcal{B}^2}}{\mathcal{B}}. \tag{1}$$

Now, assume that $\mathbf{w}$ (i.e., the least norm classifier with a margin of $1$ on $S$) is of the form $\mathbf{w}_{\text{inv}}\mathbf{x}_{\text{inv}} + w_{\text{sp}}x_{\text{sp}} + b$. First, we derive a lower bound on $|w_{\text{sp}}|$, and after that we'll show that $w_{\text{sp}} > 0$. For this we'll consider two cases, one where $|w_{\text{sp}}| \geq \frac{1}{\mathcal{B}}$ (and so we already have a lower bound) and another case where $|w_{\text{sp}}| < \frac{1}{\mathcal{B}}$. In the latter case, we will need the invariant part of the max-margin classifier, namely $\mathbf{w}_{\text{inv}}\mathbf{x}_{\text{inv}} + b$, to have to have a margin of at least $1 - |w_{\text{sp}}|\mathcal{B}$ on $S$; if it were any lesser, the contribution from the spurious component which is at most $|w_{\text{sp}}x_{\text{sp}}|$ will be unable to bump this margin up to $1$. Now, for the invariant part of the max-margin classifier to have a margin of at least $1 - |w_{\text{sp}}|\mathcal{B}$ (a non-negative quantity since $|w_{\text{sp}}| \leq 1/\mathcal{B}$) on $S$, the norm of the (invariant part of the) classifier must be at least $(1 - |w_{\text{sp}}|\mathcal{B})\|\mathbf{v}(S)\|$ (which follows from the definition of $\mathbf{v}(S)$). This implies,

$$\|\mathbf{w}\| \geq (1 - |w_{\text{sp}}|\mathcal{B})\|\mathbf{v}(S)\|. \tag{2}$$

Combining this with Eq 1, we get $(1 - |w_{\text{sp}}|\mathcal{B})\|\mathbf{v}(S)\| \leq \frac{\sqrt{1+4\|\tilde{\mathbf{v}}(S_{\min})\|^2\mathcal{B}^2}}{\mathcal{B}}$. Rearranging this gives us the bound that $\mathcal{B}|w_{\text{sp}}| \geq 1 - 2\sqrt{\frac{\|\tilde{\mathbf{v}}(S_{\min})\|^2}{\|\mathbf{v}(S)\|^2} + \frac{1}{4\mathcal{B}^2\|\mathbf{v}(S)\|^2}}$. Finally note that $\mathbf{v}(S)$ is the same as $\tilde{\mathbf{v}}(S)$. Hence we can interchange these terms in the final result (which we've done for readability) to arrive at $|\mathcal{B}w_{\text{sp}}| \geq 1 - 2\sqrt{\tilde{\kappa}_1 + c_1^2}$.

What remains now is to show that $w_{\text{sp}} > 0$. For this we do the same argument as above but with a slight modification. First, if $w_{\text{sp}} > 1/\mathcal{B}$, we are done. So, assume that $w_{\text{sp}} \leq 1/\mathcal{B}$. Then, we can say that the invariant part of the max-margin classifier i.e., $\mathbf{w}_{\text{inv}}\mathbf{x}_{\text{inv}} + b$, must achieve a margin of $1 - w_{\text{sp}}\mathcal{B}$ (a non-negative quantity since $w_{\text{sp}} \leq 1/\mathcal{B}$) specifically on $S_{\text{maj}}$. Then, by the definition of $\mathbf{v}(S_{\text{maj}})$, it follows that the norm of our overall max-margin classifier must be at least $(1 - w_{\text{sp}}\mathcal{B})\|\mathbf{v}(S_{\text{maj}})\|$. Again, for the overall classifier to be the max-margin classifier, we need $(1 - |w_{\text{sp}}|\mathcal{B})\|\mathbf{v}(S_{\text{maj}})\| \leq \frac{\sqrt{1+4\|\tilde{\mathbf{v}}(S_{\min})\|^2\mathcal{B}^2}}{\mathcal{B}}$, which when rearranged gives us $\mathcal{B}w_{\text{sp}} \geq 1 - \frac{\sqrt{4\mathcal{B}^2\|\tilde{\mathbf{v}}(S_{\min})\|^2+1}}{\mathcal{B}\|\tilde{\mathbf{v}}(S_{\text{maj}})\|}$. The R.H.S is at least $0$ when $\frac{1}{4}\left(1 - \frac{1}{\|\tilde{\mathbf{v}}(S_{\text{maj}})\|^2\mathcal{B}^2}\right) \geq \frac{\|\tilde{\mathbf{v}}(S_{\min})\|^2}{\|\tilde{\mathbf{v}}(S_{\text{maj}})\|^2}$ (i.e., $\sqrt{\frac{1}{4} - c_2^2} \geq \tilde{\kappa}_2$). In other words when $\tilde{\kappa}_2 \leq \sqrt{\frac{1}{4} - c_2^2}$, we have $w_{\text{sp}} \geq 0$.

**Proof of upper bound.** The spurious component of the classifier $\mathbf{w}_{\text{inv}}\mathbf{x}_{\text{inv}} + w_{\text{sp}}x_{\text{sp}} + b$ positively contributes to the margin of one of the groups (i.e., one of $S_{\min}$ and $S_{\text{maj}}$), and negatively contributes to the other group, depending on the sign of $w_{\text{sp}}$. On whichever group the spurious component negatively contributes to the margin, the invariant part of the classifier, $\mathbf{w}_{\text{inv}}\mathbf{x}_{\text{inv}} + b$, must counter this and achieve a margin of $1 + |w_{\text{sp}}|\mathcal{B}$. To manage this, we'd require $\|\mathbf{w}_{\text{inv}}\| \geq (1 + |w_{\text{sp}}|\mathcal{B})\min(\|\mathbf{v}(S_{\min})\|, \|\mathbf{v}(S_{\text{maj}})\|)$. In other words, for the overall max-margin classifier $\mathbf{w}$, we have:

$$\|\mathbf{w}\| \geq (1 + |w_{\text{sp}}|\mathcal{B})\min(\|\mathbf{v}(S_{\min})\|, \|\mathbf{v}(S_{\text{maj}})\|). \tag{3}$$

At the same time, we also know from the definition of $\mathbf{v}(S)$ that

$$\|\mathbf{w}\| \leq \|\mathbf{v}(S)\|. \tag{4}$$

Combining the above two equations, we can say $(1 + |w_{\text{sp}}|\mathcal{B})\min(\|\mathbf{v}(S_{\min})\|, \|\mathbf{v}(S_{\text{maj}})\|) \leq \|\mathbf{v}(S)\|$. Since we are given $\kappa_2 \leq 1$, it means that $\min(\|\mathbf{v}(S_{\min})\|, \|\mathbf{v}(S_{\text{maj}})\|) = \|\mathbf{v}(S_{\min})\|$, this simplifies to $(1 + |w_{\text{sp}}|\mathcal{B})\|\mathbf{v}(S_{\min})\| \leq \|\mathbf{v}(S)\|$, which when rearranged reaches the result $\mathcal{B}w_{\text{sp}} \leq \frac{1}{\kappa_1} - 1$.

To get the other upper bound here, observe that for $\mathbf{w}_{\text{inv}}\mathbf{x}_{\text{inv}} + w_{\text{sp}}x_{\text{sp}} + b$ to be the overall min-norm classifier, its $\ell_2$ norm, which is lower bounded by $|w_{\text{sp}}|$ must not be larger than the $\ell_2$ norm of $\mathbf{v}(S)$. Hence $|w_{\text{sp}}| \leq \|\mathbf{v}(S)\|$. $\square$

### B.2 PROOF OF THEOREM 2 ON FAILURE DUE TO STATISTICAL SKEWS

Below we state the full form of Theorem 2 and its proof demonstrating the effect of statistical skews in easy-to-learn tasks. After that, we'll present a more precise analysis of the same in a 2D setting in Theorem 5 (for exponential loss) and in Theorem 6 (for logistic loss).

Our result below focuses on any easy-to-learn task and on a corresponding dataset where there are no geometric skews. Specifically, we consider a dataset where the invariant features have the same empirical distribution in both the majority subset (where $x_{\text{sp}} \cdot y > 0$) and the minority subset (where $x_{\text{sp}} \cdot y < 0$). As a result, in this setting the max-margin classifier would not rely on the spurious feature. This allows us to focus on a setting where we can isolate and study the effect of statistical skews.

For the sake of convenience, we focus on the exponential loss and under infinitesimal learning rate, and a classifier initialized to the origin.

**Theorem 4.** *(full form of Theorem 2) Let $\mathbb{H}$ be the set of linear classifiers, $h(\mathbf{x}) = \mathbf{w}_{\text{inv}}\mathbf{x}_{\text{inv}} + w_{\text{sp}}x_{\text{sp}}$. Consider any task that satisfies all the constraints in Section 3.1. Consider a dataset $S$ drawn from $\mathcal{D}$ such that the empirical distribution of $\mathbf{x}_{\text{inv}}$ given $x_{\text{sp}} \cdot y > 0$ is identical to the empirical distribution of $\mathbf{x}_{\text{inv}}$ given $x_{\text{sp}} \cdot y < 0$. Let $\mathbf{w}_{\text{inv}}(t)\mathbf{x}_{\text{inv}} + w_{\text{sp}}(t)x_{\text{sp}}$ be initialized to the origin, and trained with an infinitesimal rate to minimize the exponential loss on a dataset $S$. Then, for any $(\mathbf{x}, y) \in S$, we have:*

$$\Omega\left(\frac{\ln\frac{c+p}{c+\sqrt{p(1-p)}}}{\mathcal{M}\ln(t+1)}\right) \leq \frac{w_{\text{sp}}(t)\mathcal{B}}{|\mathbf{w}_{\text{inv}}(t) \cdot \mathbf{x}_{\text{inv}}|} \leq \mathcal{O}\left(\frac{\ln\frac{p}{1-p}}{\ln(t+1)}\right)$$

*where:*

- *$p$ denotes the empirical level of spurious correlation, $p = \frac{1}{|S|}\sum_{(\mathbf{x},y)\in S}\mathbf{1}[x_{\text{sp}}\cdot y > 0]$ which without generality is assumed to satisfy $p \in [0.5, 1)$.*

- *$\mathcal{M}$ denotes the maximum value of the margin of the max-margin classifier on $S$ i.e., $\mathcal{M} = \max_{\mathbf{x}\in S}\hat{\mathbf{w}} \cdot \mathbf{x}$ where $\hat{\mathbf{w}}$ is the max-margin classifier on $S$.*

- *$c := \frac{2(2\mathcal{M}-1)}{\mathcal{B}^2}$*

*Proof.* Throughout the discussion, we'll denote $\mathbf{w}_{\text{inv}}(t)$ and $w_{\text{sp}}(t)$ as just $\mathbf{w}_{\text{inv}}$ and $w_{\text{sp}}$ for readability.

Let $S_{\text{min}}$ and $S_{\text{maj}}$ denote the subset of datapoints in $S$ where $x_{\text{sp}} \cdot y < 0$ and $x_{\text{sp}} \cdot y > 0$ respectively. Let $\hat{\mathcal{D}}_{\text{inv}}$ denote the uniform distribution over $\mathbf{x}_{\text{inv}}$ induced by drawing $\mathbf{x}$ uniformly from $S_{\text{min}}$. By the assumption of the theorem, this distribution would be the same if $\mathbf{x}$ was drawn uniformly from $S_{\text{maj}}$. Then, the loss function that is being minimized in this setting corresponds to:

$$L(\mathbf{w}_{\text{inv}}, w_{\text{sp}}) = p\mathbb{E}_{\mathbf{x}_{\text{inv}}\sim\hat{\mathcal{D}}_{\text{inv}}}\left[e^{-(\mathbf{w}_{\text{inv}}\cdot\mathbf{x}_{\text{inv}}+w_{\text{sp}}\mathcal{B})}\right] + (1-p)\mathbb{E}_{\mathbf{x}_{\text{inv}}\sim\hat{\mathcal{D}}_{\text{inv}}}\left[e^{-(\mathbf{x}_{\text{inv}}\cdot\mathbf{w}_{\text{inv}}-w_{\text{sp}}\mathcal{B})}\right],$$

where $p \in [0.5, 1)$. Here, the first term is the loss on the majority dataset (where $x_{\text{sp}} = y\mathcal{B}$) and the second term is the loss on the minority dataset (where $x_{\text{sp}} = -y\mathcal{B}$).

The update on $w_{\text{sp}}$ can be written as:

$$\dot{w}_{\text{sp}} = \mathbb{E}_{\mathbf{x}_{\text{inv}}\sim\hat{\mathcal{D}}_{\text{inv}}}\left[e^{-\mathbf{w}_{\text{inv}}\mathbf{x}_{\text{inv}}}\right] \cdot \mathcal{B} \cdot \left(pe^{-w_{\text{sp}}\mathcal{B}} - (1-p)e^{w_{\text{sp}}\mathcal{B}}\right)$$

To study the dynamics of this quantity, we first bound the value of $\mathbf{w}_{\text{inv}}(t)\mathbf{x}_{\text{inv}}$.

**Bounds on $\mathbf{w}_{\text{inv}}(t)\mathbf{x}_{\text{inv}}$** The result from Soudry et al. (2018) states that we can write $\mathbf{w}(t) = \hat{\mathbf{w}}\ln(1+t) + \rho(t)$ where $\hat{\mathbf{w}}$ is the max-margin classifier and $\rho$ is a residual vector that is bounded as $\|\rho(t)\|_2 = O(\ln\ln t)$. Since the max-margin classifier here is of the form $\hat{\mathbf{w}} = (\hat{\mathbf{w}}_{\text{inv}}, 0)$ (i.e., it only relies on the invariant feature), we can infer from this that $\mathbf{w}_{\text{inv}}(t) = \hat{\mathbf{w}}_{\text{inv}}\ln(1+t) + \rho^{\dagger}(t)$ where again $\|\rho^{\dagger}(t)\|_2 = O(\ln\ln t)$. For a sufficiently large $t$, we can

say that $\ln \ln t \ll \ln(1 + t)$. This would then imply that for all $\mathbf{x} \in S$, $|\mathbf{w}_{\text{inv}}(t) \cdot \mathbf{x}_{\text{inv}}| \in [0.5\hat{\mathbf{w}}_{\text{inv}}\mathbf{x}_{\text{inv}}(t) \ln(1 + t), 2\hat{\mathbf{w}}_{\text{inv}}\mathbf{x}_{\text{inv}}(t) \ln(1 + t)]$. Since the max-margin classifier has a margin between 1 and $\mathcal{M}$ on the training data, this implies that, for a sufficiently large $t$ and for all $\mathbf{x} \in S$:

$$|\mathbf{w}_{\text{inv}}(t) \cdot \mathbf{x}_{\text{inv}}| \in [0.5 \ln(1 + t), 2\mathcal{M} \ln(1 + t)].$$

Next, we bound the dynamics of $w_{\text{sp}}$.

**Upper bound on $w_{\text{sp}}$.** To upper bound $w_{\text{sp}}$, we first note that $\dot{w}_{\text{sp}} = 0$ only when $w_{\text{sp}} = \frac{1}{2\mathcal{B}} \ln \frac{p}{1-p}$. Furthermore, $\dot{w}_{\text{sp}}$ is a decreasing function in $w_{\text{sp}}$. Hence, for any value of $w_{\text{sp}}$ that is less than $\frac{1}{2\mathcal{B}} \ln \frac{p}{1-p}$, $\dot{w}_{\text{sp}} \geq 0$ and for any that is greater than this value, $\dot{w}_{\text{sp}} \leq 0$. So, we can conclude that when the system is initialized at 0, it can never cross the point $w_{\text{sp}} = \frac{1}{2\mathcal{B}} \ln \frac{p}{1-p}$. In other words, for all $t$, $w_{\text{sp}}(t) \leq \frac{1}{2\mathcal{B}} \ln \frac{p}{1-p}$. Combining this with the lower bound on $\mathbf{w}_{\text{inv}}(t)$, we get the desired result.

**Lower bound on $w_{\text{sp}}$.** We lower bound $\dot{w}_{\text{sp}}$ via the upper bound on $w_{\text{sp}}$ as:

$$\dot{w}_{\text{sp}} \geq \mathbb{E}_{\mathbf{x}_{\text{inv}} \sim \hat{\mathcal{D}}_{\text{inv}}} \left[ e^{-\mathbf{w}_{\text{inv}}\mathbf{x}_{\text{inv}}} \right] \cdot \mathcal{B} \cdot \left( pe^{-w_{\text{sp}}\mathcal{B}} - (1 - p)\frac{p}{1 - p} \right)$$

$$= \mathbb{E}_{\mathbf{x}_{\text{inv}} \sim \hat{\mathcal{D}}_{\text{inv}}} \left[ e^{-\mathbf{w}_{\text{inv}}\mathbf{x}_{\text{inv}}} \right] \cdot \mathcal{B} \cdot \left( pe^{-w_{\text{sp}}\mathcal{B}} - \sqrt{p(1 - p)} \right).$$

Next, since we have that for all $\mathbf{x} \in S$, $|\mathbf{w}_{\text{inv}} \cdot \mathbf{x}_{\text{inv}}| \leq 2\mathcal{M} \ln(t + 1)$:

$$\dot{w}_{\text{sp}} \geq \frac{1}{(t + 1)^{2\mathcal{M}}} \mathcal{B} \cdot \left( pe^{-w_{\text{sp}}\mathcal{B}} - \sqrt{p(1 - p)} \right).$$

Rearranging this and integrating, we get:

$$\int_0^{w_{\text{sp}}} \frac{1}{pe^{-w_{\text{sp}}\mathcal{B}} - \sqrt{p(1 - p)}} dw_{\text{sp}} \geq \int_0^t \mathcal{B} \frac{1}{(1 + t)^{2\mathcal{M}}} dt,$$

(Since $2\mathcal{M} \geq 2$, we can integrate the right hand side as below)

$$-\frac{\ln(p - e^{w_{\text{sp}}\mathcal{B}}\sqrt{p(1 - p)})}{\mathcal{B}\sqrt{p(1 - p)}} + \frac{\ln(p - \sqrt{p(1 - p)})}{\mathcal{B}\sqrt{p(1 - p)}} \geq \frac{\mathcal{B}}{2\mathcal{M} - 1} \left( 1 - \frac{1}{(1 + t)^{2\mathcal{M} - 1}} \right),$$

since for a sufficiently large $t$, the final paranthesis involving $t$ will at least be half,

$$\ln \left( \frac{\sqrt{\frac{p}{1-p}} - 1}{\sqrt{\frac{p}{1-p}} - e^{w_{\text{sp}}\mathcal{B}}} \right) \geq \frac{1}{2} \frac{\sqrt{p(1 - p)}\mathcal{B}^2}{2\mathcal{M} - 1},$$

we can further lower bound the right hand side by applying the inequality $x \geq \ln(x + 1)$ for positive $x$,

$$\ln \left( \frac{\sqrt{\frac{p}{1-p}} - 1}{\sqrt{\frac{p}{1-p}} - e^{w_{\text{sp}}\mathcal{B}}} \right) \geq \ln \left( 1 + \frac{1}{2} \frac{\sqrt{p(1 - p)}\mathcal{B}^2}{2\mathcal{M} - 1} \right).$$

Taking exponents on both sides and rearranging,

$$e^{w_{\text{sp}}\mathcal{B}} \geq \sqrt{\frac{p}{1 - p}} - \frac{\sqrt{\frac{p}{1-p}} - 1}{1 + \frac{1}{2}\frac{\sqrt{p(1-p)}\mathcal{B}^2}{(2\mathcal{M}-1)}}$$

$$w_{\text{sp}} \geq \frac{1}{\mathcal{B}} \ln \frac{\frac{2(2\mathcal{M}-1)}{\mathcal{B}^2} + p}{\frac{2(2\mathcal{M}-1)}{\mathcal{B}^2} + \sqrt{p(1 - p)}}.$$

Combining this with the upper bound on $w_{\text{inv}}(t)$, we get the lower bound on $w_{\text{sp}}(t)/w_{\text{inv}}(t)$.

$\square$

### B.3   PRECISE ANALYSIS OF STATISTICAL SKEWS FOR A 2D SETTING UNDER EXPONENTIAL LOSS

We now consider the 2D dataset $\mathcal{D}_{\text{2-dim}}$ considered in the main paper, with the spurious feature scale set as $\mathcal{B} = 1$, and provide a more precise analysis of the dynamics under exponential loss. This analysis is provided for the sake of completeness as the proof is self-contained and does not rely on the result of Soudry et al. (2018); Ji & Telgarsky (2018). In the next section, we perform a similar analysis for logistic loss.

**Theorem 5.** *Under the exponential loss with infinitesimal learning rate, a linear classifier* $w_{\text{inv}}(t)x_{\text{inv}} + w_{\text{sp}}(t)x_{\text{sp}}$ *initialized to the origin and trained on $\mathcal{D}_{\text{2-dim}}$ with $\mathcal{B} = 1$ satisfies:*

$$\frac{\ln\left((1+2p)/(3-2p)\right)}{\ln(1 + 3\max(t, 1))} \leq \frac{w_{\text{sp}}(t)}{w_{\text{inv}}(t)} \leq \frac{\ln\left(p/(1-p)\right)}{\ln(1 + 2t)}, \quad \text{where } p := Pr_{\mathcal{D}_{\text{2-dim}}}[x_{\text{sp}} \cdot y > 0] \in [0.5, 1].$$

*Proof.* Throughout the proof, we'll drop the argument $t$ from $w_{\text{inv}}(t)$ and $w_{\text{sp}}(t)$ for convenience.

The loss function that is being minimized in this setting corresponds to:

$$L(w_{\text{inv}}, w_{\text{sp}}) = pe^{-(w_{\text{inv}}+w_{\text{sp}})} + (1-p)e^{-(w_{\text{inv}}-w_{\text{sp}})},$$

where $p \geq 0.5$. Here, the first term is the loss on the majority dataset (where $x_{\text{sp}} = y\mathcal{B}$) and the second term is the loss on the minority dataset (where $x_{\text{sp}} = -y\mathcal{B}$).

Now the updates on $w_{\text{inv}}$ and $w_{\text{sp}}$ are given by:

$$\dot{w}_{\text{inv}} = pe^{-(w_{\text{inv}}+w_{\text{sp}})} + (1-p)e^{-(w_{\text{inv}}-w_{\text{sp}})}$$
$$\dot{w}_{\text{sp}} = pe^{-(w_{\text{inv}}+w_{\text{sp}})} - (1-p)e^{-(w_{\text{inv}}-w_{\text{sp}})},$$

which means:

$$\frac{d(w_{\text{inv}} + w_{\text{sp}})}{dt} = 2pe^{-(w_{\text{inv}}+w_{\text{sp}})}$$
$$\frac{d(w_{\text{inv}} - w_{\text{sp}})}{dt} = 2(1-p)e^{-(w_{\text{inv}}+w_{\text{sp}})}$$

Thus, by rearranging and integrating we get:

$$w_{\text{inv}} + w_{\text{sp}} = \ln(1 + 2pt)$$
$$w_{\text{inv}} - w_{\text{sp}} = \ln(1 + 2(1-p)t)$$
$$w_{\text{inv}} = 0.5(\ln(1 + 2pt) + \ln(1 + 2(1-p)t))$$
$$w_{\text{sp}} = 0.5(\ln(1 + 2pt) - \ln(1 + 2(1-p)t)).$$

Now let us define $\beta(t) = w_{\text{sp}}/w_{\text{inv}}$:

$$\beta(t) := \frac{w_{\text{sp}}(t)}{w_{\text{inv}}(t)} = \frac{\ln(1 + 2pt) - \ln(1 + 2(1-p)t)}{\ln(1 + 2pt) + \ln(1 + 2(1-p)t)}. \tag{5}$$

To bound this quantity, we'll consider two cases, $t \geq 1$ and $t < 1$. First let us consider $t \geq 1$. We begin by noting that the numerator $w_{\text{sp}}(t)$ is increasing with time $t$. This is because,

$$w_{\text{sp}}(t) = \ln\frac{1 + 2pt}{1 + 2(1-p)t}$$

$$= \ln\left(1 + \frac{2(2p-1)t}{1 + 2(1-p)t}\right)$$

$$= \ln\left(1 + \frac{2(2p-1)}{\frac{1}{t} + 2(1-p)}\right).$$

Here, the term $\frac{2(2p-1)}{\frac{1}{t}+2(1-p)}$ is increasing with $t$ due to the fact that the numerator is non-negative ($p \geq 0.5$) and the denominator is decreasing with $t$. So, given that $w_{\mathrm{sp}}(t)$ is increasing, we can say that for all $t \geq 1$:

$$\beta(t) \geq \frac{w_{\mathrm{sp}}(1)}{w_{\mathrm{inv}}(t)} = \frac{\ln\frac{1+2p}{3-2p}}{\ln(1+2(1-p)t) + \ln(1+2pt)} \geq \frac{\ln\frac{1+2p}{3-2p}}{\ln(1+3t)}.$$

Here we have used the fact that the denominator can be upper bounded as $\ln(1+2(1-p)t)+\ln(1+2pt) \leq \ln(1+2t+4(1-p)pt) \leq \ln(1+3t)$.

Now, for any $t \leq 1$, we can show that $\beta(t) \geq \beta(1) = \frac{\ln\frac{1+2p}{3-2p}}{\ln(3+4(p-p^2))} \geq \frac{\ln\frac{1+2p}{3-2p}}{\ln 4}$. This follows if we can show that $\beta(t)$ is decreasing for $t \geq 0$. Taking its derivative with respect to time, we get:

$$\dot{\beta} = \frac{(\ln(1+2pt) + \ln(1+2(1-p)t))\left(\frac{2p}{1+2pt} - \frac{2(1-p)}{(1+2(1-p)t)}\right)}{(\ln(1+2pt) + \ln(1+2(1-p)t))^2}$$

$$- \frac{(\ln(1+2pt) - \ln(1+2(1-p)t))\left(\frac{2p}{(1+2pt)} + \frac{2(1-p)}{1+2(1-p)t}\right)}{(\ln(1+2pt) + \ln(1+2(1-p)t))^2}$$

$$= 2 \cdot \frac{\ln(1+2(1-p)t)\frac{2p}{1+2pt} - \ln(1+2pt)\frac{2(1-p)}{1+2(1-p)t}}{(\ln(1+2pt) + \ln(1+2(1-p)t))^2}$$

$$= 2 \cdot \frac{\ln(1+2(1-p)t)\frac{1}{\frac{1}{2p}+t} - \ln(1+2pt)\frac{1}{\frac{1}{2(1-p)}+t}}{(\ln(1+2pt) + \ln(1+2(1-p)t))^2}$$

The sign of the above quantity is equal to the sign of:

$$\ln(1+2(1-p)t)\left(\frac{1}{2(1-p)} + t\right) - \ln(1+2pt)\left(\frac{1}{2p} + t\right)$$

$$= \underbrace{\ln\left(\frac{1}{2(1-p)} + t\right)\left(\frac{1}{2(1-p)} + t\right) + \ln(\frac{1}{2(1-p)})\left(\frac{1}{2(1-p)} + t\right)}_{:= f\left(\frac{1}{2(1-p)}\right)}$$

$$- \underbrace{\ln\left(\frac{1}{2p} + t\right)\left(\frac{1}{2p} + t\right) - \ln\frac{1}{2p}\left(\frac{1}{2p} + t\right)}_{:= f\left(\frac{1}{2p}\right)}$$

Now, we show that $f(x) = (x+t)\ln(x+t) - (x+t)\ln x = (x+t)\ln\left(1 + \frac{t}{x}\right)$ is a non-increasing function:

$$f'(x) = \ln\left(1 + \frac{t}{x}\right) + \frac{x+t}{1 + \frac{t}{x}} \cdot \frac{-t}{x^2}$$

$$= \ln\left(1 + \frac{t}{x}\right) - \frac{t}{x}$$

$$\le \frac{t}{x} - \frac{t}{x} \le 0.$$

Now since $p \ge 0.5$, and $f$ is non-increasing, $f\left(\frac{1}{2(1-p)}\right) - f\left(\frac{1}{2p}\right) \le 0$. Subsequently, $\dot{\beta} \le 0$. Therefore, $\beta(t) \ge \beta(1)$ for any $t \in [0, 1]$.

**Upper bound.** For an upper bound on $\beta(t)$, we note that since $w_{\text{sp}}(t)$ is always increasing $w_{\text{sp}}(t) \le \lim_{t\to\infty} w_{\text{sp}}(t) = \ln\left(\frac{p}{1-p}\right)$. On the other hand $w_{\text{inv}}(t) = \ln(1 + 2t + 4p(1-p)t^2) \ge \ln(1 + 2t)$. Combining these inequalities, we get:

$$\beta(t) \le \frac{\ln\left(\frac{1}{p} - 1\right)}{\ln(1 + 2t)}.$$

$\square$

## B.4 ANALYSIS OF STATISTICAL SKEWS FOR A 2D SETTING UNDER LOGISTIC LOSS

While the Theorem 2 and Theorem 5 were concerned with the exponential losses, as noted in Soudry et al. (2018), the dynamics under logistic loss are similar (although harder to analyze). For the sake of completeness, we show similar results for logistic loss in the same 2D setting as Theorem 5.

**Theorem 6.** *Under the logistic loss with infinitesimal learning rate, a linear classifier $w_{\text{inv}}(t)x_{\text{inv}} + w_{\text{sp}}(t)x_{\text{sp}}$ initialized to the origin and trained on $\mathcal{D}_{2\text{-dim}}$ with $\mathcal{B} = 1$ satisfies for a sufficiently large $t$ (where $p := Pr_{\mathcal{D}_{2\text{-dim}}}[x_{\text{sp}} \cdot y > 0] \in [0.5, 1]$):*

$$\min\left(1, \frac{\frac{1}{2}\ln\left(\frac{2}{3-2p}\right)}{\ln(t+1)}\right) \le \frac{w_{\text{sp}}(t)}{w_{\text{inv}}(t)} \le \frac{\frac{1}{2}\ln\frac{1-p}{p}}{\ln(0.5t+1)}.$$

*Proof.* Here, the loss function is of the form:

$$L(w_{\text{inv}}, w_{\text{sp}}) = p\log(1 + e^{-(w_{\text{inv}}+w_{\text{sp}})}) + (1-p)\log(1 + e^{-(w_{\text{inv}}-w_{\text{sp}})})$$

where $p \ge 0.5$. Now the updates on $w_{\text{inv}}$ and $w_{\text{sp}}$ are:

$$\dot{w}_{\text{inv}} = p\frac{e^{-(w_{\text{inv}}+w_{\text{sp}})}}{1 + e^{-(w_{\text{inv}}+w_{\text{sp}})}} + (1-p)\frac{e^{-(w_{\text{inv}}-w_{\text{sp}})}}{1 + e^{-(w_{\text{inv}}-w_{\text{sp}})}}$$

$$\dot{w}_{\text{sp}} = p\frac{e^{-(w_{\text{inv}}+w_{\text{sp}})}}{1 + e^{-(w_{\text{inv}}+w_{\text{sp}})}} - (1-p)\frac{e^{-(w_{\text{inv}}-w_{\text{sp}})}}{1 + e^{-(w_{\text{inv}}-w_{\text{sp}})}},$$

which means:

$$\frac{d(w_{\text{inv}} + w_{\text{sp}})}{dt} = 2p\frac{e^{-(w_{\text{inv}}+w_{\text{sp}})}}{1 + e^{-(w_{\text{inv}}+w_{\text{sp}})}} = 2p\frac{1}{1 + e^{(w_{\text{inv}}+w_{\text{sp}})}}$$

$$\frac{d(w_{\text{inv}} - w_{\text{sp}})}{dt} = 2(1-p)\frac{e^{-(w_{\text{inv}}-w_{\text{sp}})}}{1 + e^{-(w_{\text{inv}}-w_{\text{sp}})}} = 2(1-p)\frac{1}{1 + e^{(w_{\text{inv}}-w_{\text{sp}})}}.$$

Solving for this, we get:

$$w_{\text{inv}} + w_{\text{sp}} + e^{w_{\text{inv}}+w_{\text{sp}}} = 2pt + 1 \tag{6}$$

$$w_{\text{inv}} - w_{\text{sp}} + e^{w_{\text{inv}} - w_{\text{sp}}} = 2(1-p)t + 1. \tag{7}$$

We first derive some useful inequalities.

First, we argue that forall $t$,

$$w_{\text{sp}}(t) \geq 0. \tag{8}$$

This is because at the point where $w_{\text{sp}}(t) = 0$, $\dot{w}_{\text{sp}}(t) \geq \frac{2p-1}{1+e^{w_{\text{inv}}}} \geq 0$ (since $p \geq 0.5$). Hence, the system can never reach values of $w_{\text{sp}} < 0$.

Next, we have for all $t$,

$$w_{\text{inv}}(t) \in [0, \ln(t+1)]. \tag{9}$$

We can show this by summing up Eq 6 and 7

$$2w_{\text{inv}} + e^{w_{\text{inv}} + w_{\text{sp}}} + e^{w_{\text{inv}} - w_{\text{sp}}} = 2t + 2$$
$$\implies \quad 2w_{\text{inv}} + 2\sqrt{e^{w_{\text{inv}} + w_{\text{sp}}} \cdot e^{w_{\text{inv}} - w_{\text{sp}}}} \leq 2t + 2$$
$$\implies \quad 2w_{\text{inv}} + 2e^{w_{\text{inv}}} \leq 2t + 2$$

and since $\dot{w}_{\text{inv}} \geq 0$, and $w_{\text{inv}}(t) = 0$, $w_{\text{inv}}(t) \geq 0$,

$$2e^{w_{\text{inv}}} \leq 2t + 2$$

Next, we show:

$$w_{\text{sp}}(t) \leq \frac{1}{2} \ln \frac{2(1-p)t + 1}{2pt + 1} \leq \frac{1}{2} \ln \frac{(1-p)}{p}. \tag{10}$$

To show this, we divide Eq 6 by Eq 7, to get:

$$\frac{w_{\text{inv}} + w_{\text{sp}} + e^{w_{\text{inv}} + w_{\text{sp}}}}{w_{\text{inv}} - w_{\text{sp}} + e^{w_{\text{inv}} - w_{\text{sp}}}} = \frac{2(1-p)t + 1}{2pt + 1}$$
$$\implies \quad (2(2p-1)t)w_{\text{inv}} + (2(2p-1)t)w_{\text{sp}} + (2pt+1)e^{w_{\text{inv}} + w_{\text{sp}}} = e^{w_{\text{inv}} - (2(1-p)t+1)w_{\text{sp}}}$$

since by $p \geq 0.5$, Eq 8 and Eq 9 the first two terms are positive,

$$\implies \quad (2pt+1)e^{w_{\text{inv}} + w_{\text{sp}}} \leq (2(1-p)t + 1)e^{w_{\text{inv}} - w_{\text{sp}}}$$
$$\implies \quad e^{2w_{\text{sp}}} \leq \frac{2(1-p)t + 1}{2pt + 1}.$$

This proves the first inequality. The second inequality follows from the fact that $\frac{2(1-p)t+1}{2pt+1}$ is increasing with $t$ so applying $\lim t \to \infty$ gives us an upper bound.

Finally, we rewrite Equation 6 and Equation 7 to get:

$$w_{\text{inv}} + w_{\text{sp}} = \ln(2pt + 1 - (w_{\text{inv}} + w_{\text{sp}})) \tag{11}$$
$$w_{\text{inv}} - w_{\text{sp}} = \ln(2(1-p)t + 1 - (w_{\text{inv}} - w_{\text{sp}}). \tag{12}$$

Adding and subtracting these, we get a different form for the dynamics of these quantities:

$$w_{\text{inv}} = 0.5(\ln(2pt + 1 - (w_{\text{inv}} + w_{\text{sp}})) + \ln(2(1-p)t + 1 - (w_{\text{inv}} - w_{\text{sp}})) \tag{13}$$
$$w_{\text{sp}} = 0.5(\ln(2pt + 1 - (w_{\text{inv}} + w_{\text{sp}})) - \ln(2(1-p)t + 1 - (w_{\text{inv}} - w_{\text{sp}})). \tag{14}$$

**Lower bound.** To prove a lower bound on $w_{sp}(t)/w_{inv}(t)$, we'll first lower bound $w_{sp}$. Observe that:

$$w_{sp}(t) = \frac{1}{2} \ln \frac{2pt + 1 - (w_{inv} + w_{sp})}{2(1-p)t + 1 - (w_{inv} - w_{sp})}$$
$$= \frac{1}{2} \ln \left( 1 + \frac{2(2p-1)t - 2w_{sp}}{2(1-p)t + 1 - (w_{inv} - w_{sp})} \right)$$

Now, since $w_{sp}$ is upper bounded by a constant (Eq 10), for sufficiently large $t$, the numerator of the second term inside the $\ln$ will be positive, and can be lower bounded by $(2p-1)t$. Then, let us consider two scenarios. Either that $w_{sp} > w_{inv}$, in which case we already have a lower bound on $\beta(t)$, or that $w_{sp} \le w_{inv}$. In the latter case, we can lower bound the above as:

$$w_{sp}(t) \ge \frac{1}{2} \ln \left( 1 + \frac{(2p-1)t}{2(1-p)t + 1} \right)$$

Since the right hand side is an increasing in $t$, we can say that for sufficiently large $t \ge 1$,

$$w_{sp}(t) \ge \frac{1}{2} \ln \left( 1 + \frac{(2p-1)}{2(1-p) + 1} \right)$$
$$\ge \frac{1}{2} \ln \left( \frac{2}{3-2p} \right).$$

Combining this with Eq 9 we get for sufficiently large $t$, either

$$\frac{w_{sp}(t)}{w_{inv}(t)} \ge \frac{\frac{1}{2} \ln \left( \frac{2}{3-2p} \right)}{\ln(t+1)}.$$

or $\frac{w_{sp}(t)}{w_{inv}(t)} \ge 1$.

**Upper bound.** To upper bound $w_{sp}(t)/w_{inv}(t)$, we'll lower bound $w_{inv}(t)$:

$$w_{inv} = 0.5(\ln(2pt + 1 - (w_{inv} + w_{sp})) + \ln(2(1-p)t + 1 - (w_{inv} - w_{sp}))$$

Since by Eq 9 $w_{inv}(t) \in [0, \ln(t+1)]$ and by Eq 10, $w_{sp}(t) \in [0, \frac{1}{2} \ln \frac{1-p}{p}]$, for a sufficiently large $t$, the linear terms within the $\ln$ terms dominate and so, for large $t$

$$w_{inv} \ge (\ln(0.5 \cdot 2pt + 1) = \ln(0.5t + 1)$$

Combining this with Eq 10, we get, for large $t$:

$$\frac{w_{sp}(t)}{w_{inv}(t)} \le \frac{\frac{1}{2} \ln \frac{1-p}{p}}{\ln(0.5t + 1)}.$$

$\square$

## C    MORE ON EXPERIMENTS

**Common details:** In all our MNIST-based experiments, we consider the Binary-MNIST classification task (Arjovsky et al., 2019) where the first five digits (0 to 4) need to be separated from the rest

(5 to 9). Unless specified otherwise, for this we train a fully-connected three-layered ReLU network with a width of $400$ and using SGD with learning rate $0.1$ for 50 epochs. In all our CIFAR10-based experiments, unless stated otherwise, we consider the 10-class classification problem, and train a ResNetV1 with a depth of 20 for 200 epochs. [7] All values are averaged at least over fives runs. Finally, when we describe our datasets, we'll adopt the convention that all pixels lie between 0 and 1,

## C.1 RANDOM FEATURE EXPERIMENTS FROM SECTION 3.1

For the random features based experiment on Binary MNIST in Section 3.1, we consider $50k$ random ReLU features i.e., $\mathbf{x}_{\mathrm{inv}} = \mathrm{ReLU}(W\mathbf{x}_{\mathrm{raw}})$ where $W$ is a $50k \times 784$ matrix (and so this is well overparameterized for dataset sizes upto $6400$). Each entry here is drawn from the normal distribution. We set the spurious feature support to be $\{-100, 100\}$, which is about $1/10$th the magnitude of $\|\mathbf{x}_{\mathrm{inv}}\|$. We also conduct similar experiments on a two-class CIFAR10 and report similar OoD accuracy drops in Fig 5a. Here, we use the first two classes of CIFAR10, as against grouping five classes together into each of the classes. This is because on the latter dataset, the random features representation has poor in-distribution accuracy to begin with.

## C.2 INCREASING NORM EXPERIMENTS FROM SECTION 4

The main premise behind our geometric skews argument is that as we increase the number of data-points, it requires greater norm for the model to fit those points. We verify this for random features models on Binary-MNIST and two-class CIFAR10 in Figs 5b, 5c.

While the theory is solely focused on linear classifiers, we can verify this premise intuitively for neural network classifiers. However, for neural networks, the notion of margin is not well-defined. Nevertheless, as a proxy measure, we look at how much distance the weights travel from their initialization in order to classify the dataset completely. Such plots have already been considered in Neyshabur et al. (2017); Nagarajan & Kolter (2017; 2019) (although in the completely different context of understanding why deep networks succeed at in-distribution generalization). We present similar plots for completeness.

Fig 5d shows this for Binary-MNIST on an FNN. For CIFAR10, we conduct two experiments. Fig 5f uses a ResNet with Adam and decaying learning rate. Here, we observe that the norms saturate after a point, which is because of the learning rate decay. Since this does not make a fair comparison between the geometries of larger datasets and smaller datasets, we also plot this for SGD with fixed learning rate in Fig 5e to recover the increasing norms observation. Here, sometimes the model sometimes saturates at an accuracy of $99\%$ (rather than $100\%$); in those cases, we report the value of the weight norm at the final epoch (namely, at the 200th epoch).

## C.3 BROADER EXAMPLES OF GEOMETRIC FAILURE

We elaborate on the multiple datasets we showcased in the paper as examples where ERM fails due to a geometric skew. We first discuss the two CIFAR10 datasets, and then discuss the cats vs. dogs example, and then discuss two Binary-MNIST datasets, one that is similar to the cats vs. dogs example, and another corresponding to high-dimensional spurious features.

### C.3.1 CIFAR10 EXAMPLE WITH SPURIOUSLY COLORED LINE

Here we provide more details about the CIFAR-10 dataset we presented in Sec 4 and in Fig 1c. This dataset can be thought of as an equivalent of the cow-camel classification tasks but for 10 classes. For this, we use ten different values of the spurious feature, one for each class. We argue that the failure here arises from the fact that the ResNet requires greater norms to fit larger datapoints (see Fig 5e), and so a similar argument as Theorem 1 should explain failure here.

**Dataset details.** To create the ten-valued spurious feature, we consider a vertical line passing through the middle of each channel, and also additionally the horizontal line through the first channel. Next, we let each of these four lines take a constant value of either $(0.5 \pm 0.5\mathcal{B})$ where

---

[7]Borrowing the implementation in `https://github.com/keras-team/keras/blob/master/examples/cifar10_resnet.py`, without data augmentation.

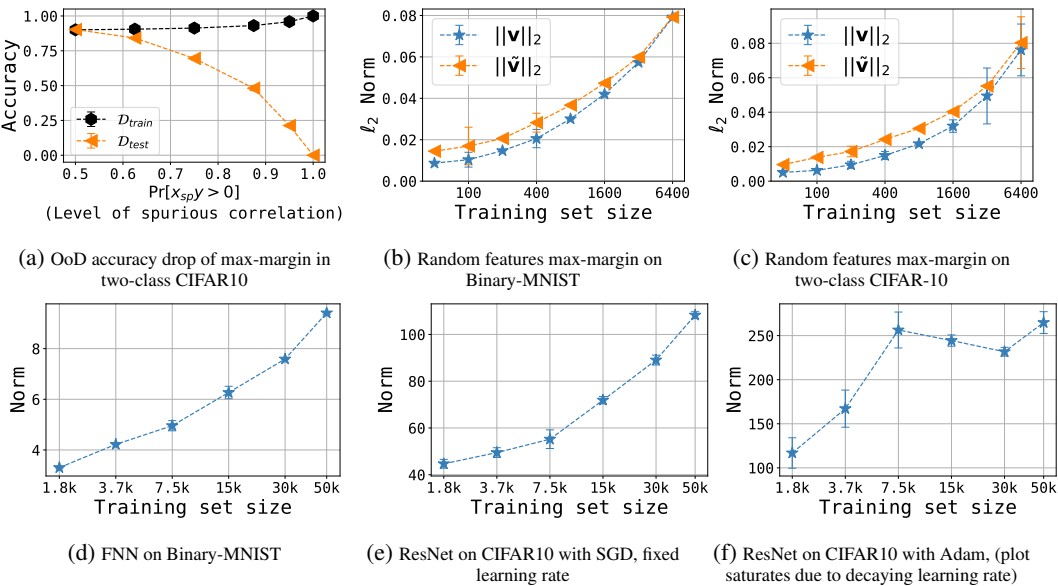

(a) OoD accuracy drop of max-margin in two-class CIFAR10

(b) Random features max-margin on Binary-MNIST

(c) Random features max-margin on two-class CIFAR-10

(d) FNN on Binary-MNIST

(e) ResNet on CIFAR10 with SGD, fixed learning rate

(f) ResNet on CIFAR10 with Adam, (plot saturates due to decaying learning rate)

Figure 5: **Validating geometric skews in MNIST and CIFAR10:** In **Fig 5a**, we show the OOD accuracy drop of a random features based max-margin model trained to classify two classes in CIFAR10. In the next few images, we demonstrate that MNIST and CIFAR10 datasets have the property that the more the datapoints in the dataset, the larger the norm required to fit them. Specifically, in **Fig 5b** and **Fig 5c**, we plot the max-margin norms of a random features representation (see App B.1 for the definitions of two plotted lines). In **Fig 5d**, **Fig 5e**, **Fig 5f**, we plot the distance from initialization of neural network models (presented for the sake of completeness).

$\mathcal{B} \in [-1, 1]$ denotes a "spurious feature scale". Since each of these lines can take two configurations, it allows us to instantiate 16 different configurations. We'll however use only 10 of these configurations, and arbitrarily fix a mapping from those configurations to the 10 classes. For convenience let us call these ten configurations $\mathbf{x}_{sp,1}, \ldots, \mathbf{x}_{sp,10}$. Then, for any datapoint in class $i$, we denote the probability of the spurious feature taking the value $\mathbf{x}_{sp,j}$, conditioned on $y$, as $p_{i,j}$.

To induce a spurious correlation, we set $p_{i,i} > 0.1$, and set all other $p_{i,j} := (1-p_{i,i})/10$. Thus, every value of the spurious feature $\mathbf{x}_{sp,j}$ is most likely to occur with its corresponding class $j$. Finally, note that to incorporate the spurious pixel, we zero out the original pixels in the image, and replace them with the spurious pixels.

For the observation reported in Fig 1c, we use the value of $\mathcal{B} = 0.5$ during training and testing. We set $p_{i,i} = 0.5$ for all classes. This means that on 50% of the data the spurious feature is aligned with the class (we call this the 'Majority' group). On the remaining 50% data, the spurious feature takes one of the other 9 values at random (we call this the 'Minority' group).

### C.3.2 CIFAR10 EXAMPLE WITH A LINE IN THE THIRD CHANNEL

Here, we elaborate on the discussion regarding the dataset in Fig 2d. In this dataset, we add a line to the last channel of CIFAR10 (regardless of the label), and vary its brightness during testing. We argue that one way to understand the failure is via the fact that the "linear mapping" Constraint 5 is broken. In particular, if we imagine that each channel contains the same invariant feature $\mathbf{x}_{inv}$ (which is almost the case as can be seen in Fig 7), then for simplicity, we can imagine this dataset to be of the form $(\mathbf{x}_{inv}, \mathbf{x}_{inv} + \mathbf{x}_{sp})$ i.e., $\mathbf{x}_{inv}$ and $\mathbf{x}_{sp}$ are not orthogonal to each other. In this scenario, the second co-ordinate can still be fully predictive of the label, and therefore the max-margin classifier would rely on both the first and the second co-ordinate to maximize its margins e.g., $\mathbf{w}_1 \cdot \mathbf{x}_{inv} + \mathbf{w}_2(\mathbf{x}_{inv} + \mathbf{x}_{sp})) + b$ where $\mathbf{w}_1, \mathbf{w}_2 \neq 0$. Crucially, since the classifier has not quite disentangled $\mathbf{x}_{sp}$ from the invariant part of the second channel, this makes the classifier vulnerable to test-time shifts on $\mathbf{x}_{sp}$. In Sec A we detail this under the failure mode of Constraint 5, and visualize this failure in Fig 4c, and also connect it to Theorem 1.

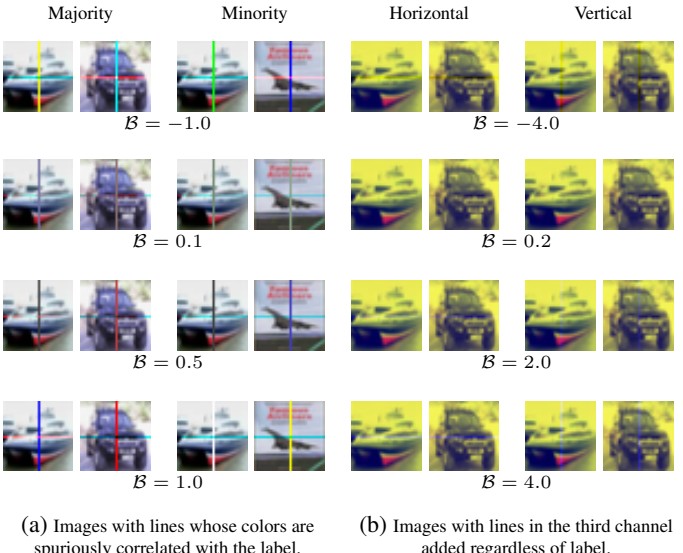

(a) Images with lines whose colors are spuriously correlated with the label.

(b) Images with lines in the third channel added regardless of label.

Figure 6: **Our CIFAR10 examples**: In **Fig 6a** we visualize the dataset discussed in App C.3.1. Each row corresponds to a different value of the "scale" of the spurious feature. The left two images correspond to datapoints where the spurious feature maps to the corresponding value for that label. The right two images correspond to datapoints where the spurious feature maps to one of the $9$ other values. In **Fig 6b**, we visualize the dataset discussed in App C.3.2. The left two images correspond to adding a horizontal line to the last channel, and the right corresponds to a vertical line.

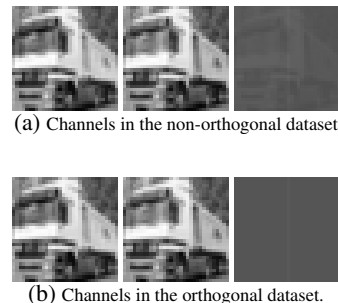

(a) Channels in the non-orthogonal dataset

(b) Channels in the orthogonal dataset.

Figure 7: **The channels in the CIFAR-10 dataset from Section C.3.2**: In the top image, is our main dataset, where one can see that the third channel has a faint copy of the original "invariant feature" and an extra vertical line added onto it. In the bottom image, is the control dataset where the last channel only contains the vertical line (thus making the spurious feature orthogonal to the invariant features).

**Dataset details.** In Fig 6b, we visualize this dataset for multiple values of a spurious feature scale parameter, $\mathcal{B} \in [-4, 4]$. In particular, we take the last channel of CIFAR10 and add $\mathcal{B}$ to the middle line of the channel. Since this can result in negative pixels, we add a value of $4$ to all pixels in the third channel, and then divide all those pixels by a value of $1 + 8$ so that they lie in between $0$ and $1$. (This normalization is the reason the color of the images differ from the original CIFAR10 dataset; as such this normalization is not crucial to our discussion.)

**More experimental results.** We run two kinds of experiments: one where we add only a vertical line to all images, and another where we add a horizontal line to $50\%$ of the images (essentially simulating data from two different domains). We also run experiments for two different values of $\mathcal{B}$ during training, $0.2$ and $1.0$. As a "control" experiment, we completely fade out the original CIFAR image in the third channel. Then, according to our explanation, the model should not fail in this setting as data is of the form $(\mathbf{x}_{\text{inv}}, \mathbf{x}_{\text{sp}})$ i.e., the two features are orthogonally decomposed. We summarize the key observations from these experiments (plotted in Fig 8) here:

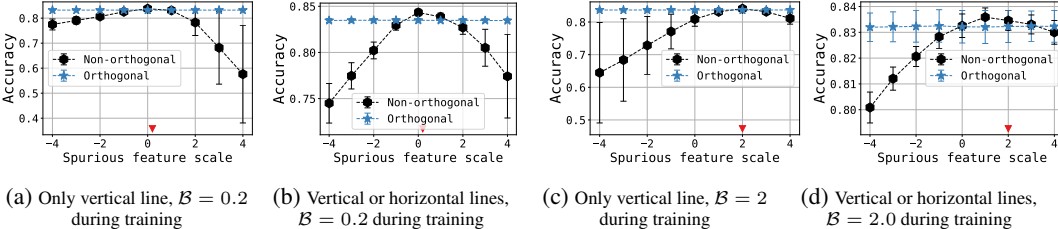

(a) Only vertical line, $\mathcal{B} = 0.2$ during training

(b) Vertical or horizontal lines, $\mathcal{B} = 0.2$ during training

(c) Only vertical line, $\mathcal{B} = 2$ during training

(d) Vertical or horizontal lines, $\mathcal{B} = 2.0$ during training

Figure 8: **More experiments on CIFAR10 example from App C.3.2**: Here the red triangle corresponds to the value of the scale of spurious feature during training. 'Non-orthogonal' corresponds to our main setting, and 'Orthogonal' corresponds to the control setting where the original image in the third channel is zeroed out.

1. As predicted, we observe that when trained on the "orthogonal" dataset (where the original CI-FAR10 image is zerod out in the third channel), the OoD performance of the neural network is unaffected. This provides evidence for our explanation of failure in the "non-orthogonal" dataset.
2. We observe that training on ERM on the "multidomain dataset" (Figs 8b, 8d) that has both horizontal and vertical lines, makes it more robust to test-time shifts compared to the dataset with purely vertical lines (Figs 8a, 8c).
3. Remarkably, even though the third channel has only a very faint copy of the image (Fig 7), the classifier still learns a significant enough weight on the channel that makes it susceptible to shifts in the middle line.
4. We note that introducing a different kind of test-time shift such as a Gaussian shift, is not powerful enough to cause failure since such a shift does not align with the weights learned, $\mathbf{w}_2$. However, shifts such as the line in this case, are more likely to be aligned with the classifier's weights, and hence cause a drop in the accuracy.

### C.3.3 CATS VS. DOGS EXAMPLE WITH COLORS INDEPENDENT OF LABEL

Recall that the dataset from Fig 1d consists of a scenario where the images of cats vs. dogs (Elson et al., 2007) are colored independently of the label. To generate this dataset, we set the first channel to be zero. Then, for a particular color, we pick a value $\mathcal{B} \in [-1, 1]$, and then set the second channel of every image to be $0.5 \cdot (1 - \mathcal{B})$ times the original first channel image, and the second channel to be $0.5 \cdot (1 + \mathcal{B})$ times the original first channel image. For the blueish images, we set $\mathcal{B} = 0.90$ and for the greenish images, we set $\mathcal{B} = -0.90$ (hence both these kinds of images have non-zero pixels in both the green and blue channels). We visualize these images in Fig 9.

Then, on the training distribution, we randomly select $p$ fraction of the data to be bluish and $1 - p$ fraction to be greenish as described above. On the testing distribution, we force all datapoints to be greenish. Finally, note that we randomly split the original cats vs. dogs dataset into 18000 points for training and use the remaining 5262 datapoints for testing/validation. In Fig 1d, we set $1 - p = 0.001$, so about $\approx 20$ greenish points must be seen during training. We show more detailed results for $1 - p$ set to $0.001, 0.01$ and $0.1$ in Fig 10. We observe that the OoD failure does diminish when $1 - p = 0.1$.

**Explaining this failure via an implicit spurious correlation.** Peculiarly, even though there is no explicit visual spurious correlation between the color and the label here, we can still identify a different kind of non-visual spurious correlation. To reason about this, first observe that, if the two active channels (the blue and the green one) correspond to $(\mathbf{x}_1, \mathbf{x}_2)$, then $\mathbf{x}_1 + \mathbf{x}_2$ is a constant across all domains (and is fully informative of the label). Hence $\mathbf{x}_1 + \mathbf{x}_2$ can hence be thought of as an invariant feature. On the other hand, consider the feature $\mathbf{x}_{\text{diff}} = \mathbf{x}_1 - \mathbf{x}_2$. During training time, this feature would correspond to a version of the original image scaled by a positive factor of $2\mathcal{B}$ for most datapoints (and scaled by a negative factor of $-2\mathcal{B}$ for a minority of datapoints). A classifier that relies only on $\mathbf{x}_1 + \mathbf{x}_2$ to predict the label will work fine on our test distribution; but if it relies on $\mathbf{x}_{\text{diff}}$, it is susceptible to fail.

Now, we informally argue why an ERM-based classifier would rely on $\mathbf{x}_{\text{diff}}$. If we were to think of this in terms of a linear classifier, we can say that there must exist a weight vector $\mathbf{w}_{\text{diff}}$ such that

Class 1          Class 2

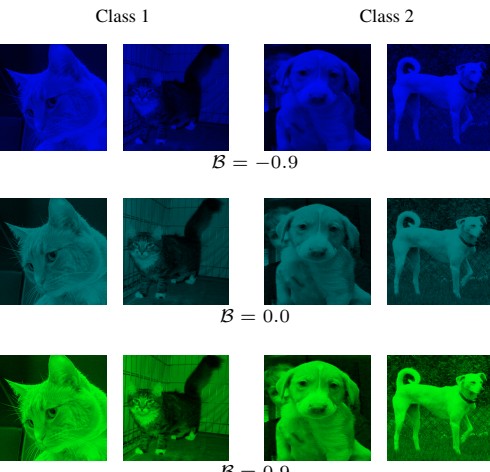

$\mathcal{B} = -0.9$

$\mathcal{B} = 0.0$

$\mathcal{B} = 0.9$

Figure 9: **Our cats vs. dogs examples**: We present the dataset from App C.3.3 for various values of $\mathcal{B}$ which determines how much of the image resides in the second channel vs. the third channel.

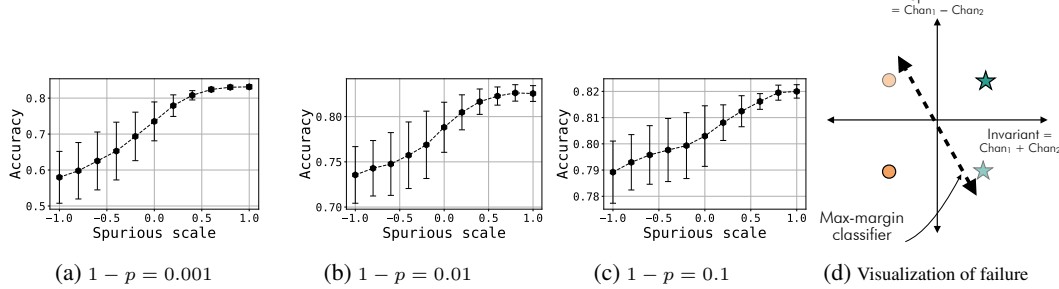

(a) $1 - p = 0.001$     (b) $1 - p = 0.01$     (c) $1 - p = 0.1$     (d) Visualization of failure

Figure 10: **Experiments on Cat vs Dogs dataset from App C.3.3**: Each of the first three plots above corresponds to a different value of $1 - p$ i.e., the proportion of the minority, greenish datapoints. We plot the OoD accuracy on various distributions for varying values of $\mathcal{B} \in [-1, 1]$. The final plot provides a visualization of these failure mode as explained in App C.3.3.

$y \cdot (\mathbf{w}_{\text{diff}} \cdot \mathbf{x}_{\text{diff}}) > 0$ for the majority of the training datapoints. Then, we can imagine a single-dimensional spurious feature that corresponds to the component of the data along this direction i.e., $x_{\text{sp}} := \mathbf{w}_{\text{diff}} \cdot \mathbf{x}_{\text{diff}}$. Notably, for a majority of the datapoints in the training set we have $x_{\text{sp}} \cdot y > 0$ and for a minority of the datapoints this feature does not necessarily correlate align with the label — let's say $x_{\text{sp}} \cdot y < 0$ for convenience. Then, this setting would effectively boil down to the geometric skew setting considered by Theorem 1. In particular, we can say that when the minority group is sufficiently small, the classifier would rely on the spurious feature $x_{\text{sp}}$ to make its classification. We visualize this in Fig 10d.

Thus, even though there is no color-label correlation in this dataset, there is still a spurious correlation, although manifest in a way that does not visually stand out. While this is one such novel kind of spurious correlation, the key message here is that we must not limit ourselves to thinking of spurious correlations as straightforward co-occurences between the label and a spurious object in the image.

**Remark 3.** It is worth distinguishing this spurious correlation with the one in the Colored MNIST dataset (Arjovsky et al., 2019). In Colored MNIST, the color is correlated with the label, and one can again think of $\mathbf{x}_{\text{diff}}$ as giving rise to the spurious feature. However, the manner in which this translates to a spurious feature is different. Here, the sign of each co-ordinate in $\mathbf{x}_{\text{diff}}$ is indicative of the true label (if it is positive, it means the image resides in the first channel, and is red, which is correlated with the label). Mathematically, if we define $\tilde{\mathbf{w}}_{\text{diff}}$ to be the vector of all ones, then we can think of $\tilde{\mathbf{w}}_{\text{diff}} \cdot \mathbf{x}_{\text{diff}}$ as a single-dimensional spurious feature here. In our case however, the vector

$\mathbf{w}_{\mathrm{diff}}$ that yields the single-dimensional spurious feature is different, and is given by the direction of separation between the two classes.

**Remark 4.** We observe that in this non-standard spurious correlation setting, we require the minority group to be much smaller than in standard spurious correlation setting (like CMNIST) to create similar levels of OoD failure. We argue that this is because the magnitude of the spurious feature $|x_{\mathrm{sp}}|$ is much smaller in this non-standard setting. Indeed, this effect of the spurious feature magnitude is captured in Theorem 3: the lower bound on the spurious component holds only when the minority group is sufficiently small to achieve $\tilde{\kappa}_2 \lesssim \sqrt{1/4 - 1/2|x_{\mathrm{sp}}|}$ i.e., when the spurious feature magnitude $|x_{\mathrm{sp}}|$ is smaller, we need the minority group to be smaller.

To see why $|x_{\mathrm{sp}}|$ differs in magnitude between the standard and non-standard spurious correlation settings, let us To see why $|x_{\mathrm{sp}}|$ differs in magnitude between the standard and non-standard spurious correlation settings, recall from the previous remark that in our setting, $|x_{\mathrm{sp}}| = |\mathbf{w}_{\mathrm{diff}} \cdot \mathbf{x}_{\mathrm{diff}}|$, while in standard spurious correlation settings $|x_{\mathrm{sp}}| = \mathbf{w}_{\mathrm{diff}} \cdot \mathbf{x}_{\mathrm{diff}}$. Intuitively, $\tilde{\mathbf{w}}_{\mathrm{diff}} \cdot \mathbf{x}_{\mathrm{diff}}$ corresponds to separating all-positive-pixel images from all-negative-pixel images, which are well-separated classes. On the other hand, $\mathbf{w}_{\mathrm{diff}} \cdot \mathbf{x}_{\mathrm{diff}}$ corresponds to separating images of one real-world class from another, which are harder to separate. Therefore, assuming that both weights vectors are scaled to unit norm, we can see that $|\mathbf{w}_{\mathrm{diff}} \cdot \mathbf{x}_{\mathrm{diff}}| \ll |\tilde{\mathbf{w}}_{\mathrm{diff}} \cdot \mathbf{x}_{\mathrm{diff}}|$

### C.3.4 BINARY-MNIST EXAMPLE WITH COLORS INDEPENDENT OF LABEL

Similar to the cats vs. dogs dataset, we also consider a Binary-MNIST dataset. To construct this, for each domain we pick a value $\mathcal{B} \in [-1, 1]$, and then set the first channel of every image to be $0.5 \cdot (1 + \mathcal{B})$ times the original MNIST image, and the second channel to be $0.5 \cdot (1 - \mathcal{B})$ times the original MNIST image. To show greater drops in OoD accuracy, we consider a stronger kind of test-time shift: during training time we set $\mathcal{B}$ to have different *positive* values so that the image is concentrated more towards the first channel; during test-time, we flip the mass completely over to the other channel by setting $\mathcal{B} = -1$.

Here again, we can visualize the dataset in terms of an invariant feature that corresponds to the sum of the two channels, and a spurious feature that corresponds to $\mathbf{w}_{\mathrm{diff}} \cdot \mathbf{x}_{\mathrm{diff}}$. The exact visualization of failure here is slightly different here since we're considering a stronger kind of shift in this setting. In particular, during training we'd have $y \cdot (\mathbf{w}_{\mathrm{diff}} \cdot \mathbf{x}_{\mathrm{diff}}) > 0$ for all the training datapoints in this setting. Thus, we can think of this as a setting with no minority datapoints in the training set (see Fig 12a). Then, as a special case of Theorem 1, we can derive a positive lower bound on the component of the classifier along $x_{\mathrm{sp}}$. However, during training time, since $\mathbf{x}_{\mathrm{diff}}$ is no longer a positively scaled version of the MNIST digits, the value of $\mathbf{w}_{\mathrm{diff}} \cdot \mathbf{x}_{\mathrm{diff}}$ would no longer be informative of the label. This leads to an overall drop in the accuracy.

**Experimental results.** We conduct two sets of experiments, one where we use only a single domain to train and another with two domains (with two unique positive values of $\mathcal{B}$). In both variations, we also consider a control setting where the data is not skewed: in the single-domain experiment, this means that we set $\mathcal{B} = 0$ (both channels have the same mass); in the two-domain experiment this means that we set $\mathcal{B}$ to be positive in one domain and negative in another. According to our explanation above, in the single-domain control setting Fig 12, since $\mathbf{x}_{\mathrm{inv}} = 0$, the classifier is likely to not rely on this direction and should be robust to test-time shifts in $\mathbf{x}_{\mathrm{diff}}$. In the two-domain control setting, since the classifier also sees negatively scaled images in $\mathbf{x}_{\mathrm{diff}}$, it would be relatively robust to such negative scaling during test-time. Indeed, we observe in Fig 12, that the performance on the non-skewed, control datasets are robust. On the other hand, on the skewed datasets, we observe greater drops in OoD accuracy when $\mathcal{B}$ is flipped, thus validating our hypothesis.

### C.3.5 BINARY-MNIST EXAMPLE WITH HIGH-DIMENSIONAL SPURIOUS FEATURES

In this dataset, we consider a two-channel MNIST setting where the second channel is a set of spurious pixels, each independently picked to be either 0 or 0.1 with probability $1 - p$ and $p$ for positively labeled points and $p$ and $1 - p$ for negatively labeled points. During training we set $p = 0.55$ and $p = 0.60$ corresponding to two training domains, and during testing, we flip this to $p = 0.0$. We visualize this dataset in Fig 11b. In Fig 13, we observe that the classifier drops to 0 accuracy during testing. To explain this failure, we can think of the sum of the pixels in the second channel as a spurious feature: since these features are independently picked, with high probability,

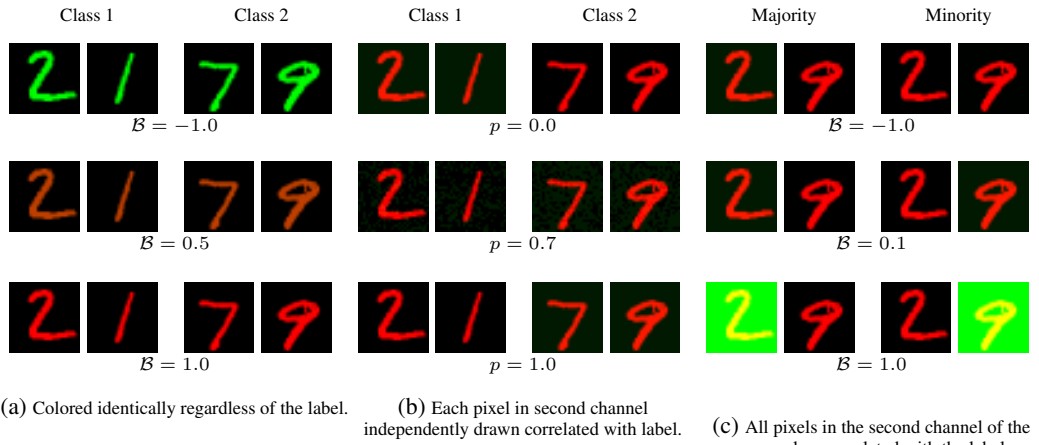

(a) Colored identically regardless of the label.

(b) Each pixel in second channel independently drawn correlated with label.

(c) All pixels in the second channel of the same value, correlated with the label.

Figure 11: **Our Binary-MNIST examples**: In **Fig 11a** we present the dataset from App C.3.4 for various values of $\mathcal{B}$ which determines how much of the image resides in the first channel. **Fig 11b** presents the dataset from App C.3.5 where the second channel pixels are individually picked to align with the label with probability $p$ (the difference is imperceptible because the pixels are either 0 or 0.1). **Fig 11c** presents the dataset for App C.4.1, where all pixels in the second channel are either 0 or $\mathcal{B}$.

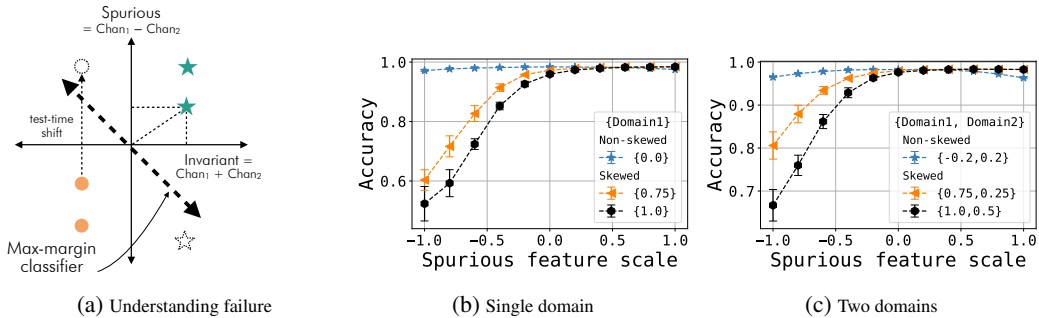

(a) Understanding failure

(b) Single domain

(c) Two domains

Figure 12: **Failure on the Binary-MNIST dataset in App C.3.4.** In **Fig 12a**, we visualize the failure observed on this dataset. Here we can think of the spurious feature as the difference between the two channels (projected along a direction $\mathbf{w}_{\text{diff}}$ where they are informative of the label) and the invariant feature as the sum of the two channels. **Fig 12b** and **Fig 12c** show the OoD performance under different shifts in the scale of the spurious feature $\mathcal{B}$. During training time this is set to the value given by 'Domain1' and/or 'Domain2'.

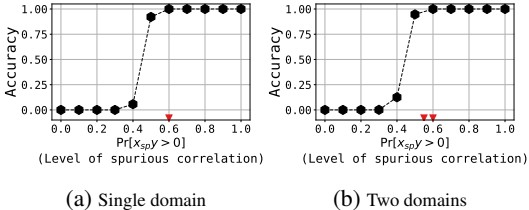

(a) Single domain

(b) Two domains

Figure 13: **Experiments for the high-dimensional spurious features dataset in App C.3.5.** The red triangles here denote the values of $p$ used during training.

their sum becomes fully informative of the label. As discussed in Section C.3.4, this boils down to the setting of Theorem 1 when there are no minority datapoints. In Appendix A, we make this argument more formal (see under the discussion of Constraint 4 in that section).

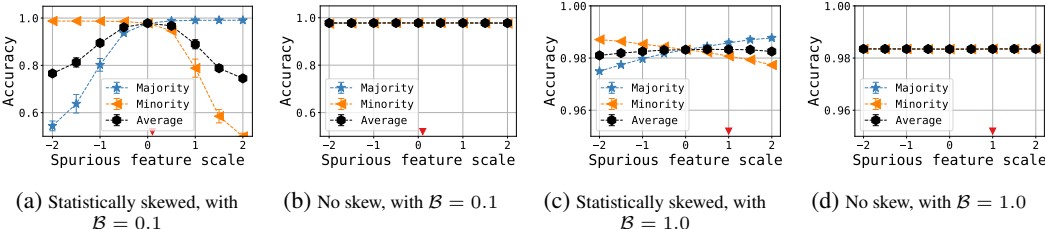

(a) Statistically skewed, with $\mathcal{B} = 0.1$

(b) No skew, with $\mathcal{B} = 0.1$

(c) Statistically skewed, with $\mathcal{B} = 1.0$

(d) No skew, with $\mathcal{B} = 1.0$

Figure 14: **Experiments validating the effect of statistical skews on the MNIST dataset in App C.4.1**. The red triangle denotes the value of $\mathcal{B}$ during training.

## C.4 EXPERIMENTS FROM SEC 5 ON STATISTICAL SKEWS

**Experiment on $\mathcal{D}_{\textbf{2-dim}}$.** For the plot in Fig 3a, we train a linear model with no bias on the logistic loss with a learning rate of $0.001$, batch size of $32$ and training set size of $2048$.

### C.4.1 BINARY-MNIST EXPERIMENTS VALIDATING THE EFFECT OF STATISTICAL SKEWS.

For this experiment, we consider a Binary-MNIST dataset where the second channel is set to be either all-0 or all-constant pixels. We visualize this dataset in Fig 11c.

To generate our control and experimental datasets, we do the following. First, we sample a set of Binary-MNIST images $S_{\text{inv}}$ and their labels from the original MNIST distribution. Next we create two datasets $S_{\text{maj}}$ and $S_{\text{min}}$ by taking each of these invariant images, and appending a spurious feature to it. More precisely, we let $S_{\text{min}} = \{((x_{\text{inv}}, x_{\text{sp}}), y) \text{ where } x_{\text{sp}} = \mathcal{B}(y+1) | x_{\text{inv}} \in S_{\text{inv}}\}$ and $S_{\text{maj}} = \{((x_{\text{inv}}, x_{\text{sp}}), y) \text{ where } x_{\text{sp}} = \mathcal{B}(-y+1) | x_{\text{inv}} \in S_{\text{inv}}\}$.

We then define a "control" dataset $S_{\text{con}} := S_{\text{maj}} \cup S_{\text{min}}$, which has a 1:1 split between the two groups of points. Next, we create an experimental "duplicated" dataset $S_{\text{exp}} := S_{\text{maj}} \cup S_{\text{min}} \cup S_{\text{dup}}$ where $S_{\text{dup}}$ is a *large* dataset consisting of datapoints randomly chosen from $S_{\text{maj}}$, thus creating a spurious correlation between the label and the spurious feature. The motivation in creating datasets this way is that neither $S_{\text{con}}$ and $S_{\text{exp}}$ have geometric skews; however $S_{\text{exp}}$ does have statistical skews, and so any difference in training on these datasets can be attributed to those skews.

**Observations.** In our experiments, we let $S_{\text{inv}}$ be a set of $30k$ datapoints, and so $S_{\text{con}}$ has $60k$ datapoints. We duplicate $S_{\text{min}}$ nine times so that $S_{\text{exp}}$ has $330k$ datapoints and has a $10 : 1$ ratio between the two groups. We consider two different settings, one where $\mathcal{B} = 0.1$ during training and $\mathcal{B} = 1.0$ during training. During testing, we report the accuracy under two kinds of shifts: shifting the value of $\mathcal{B}$, and also shifting the correlation completely to one direction (i.e., by concentrating all mass on the minority/majority group). As reported in Fig 14, the statistically skewed dataset does suffer more during test-time when compared to the unskewed dataset.

### C.4.2 CIFAR10 EXPERIMENTS VALIDATING THE EFFECT OF STATISTICAL SKEWS.

For this experiment, we consider the same CIFAR-10 dataset that we design in Section C.3.1 i.e., we introduce lines in the dataset, that can take 10 different color configurations each corresponding to one of the 10 different classes. Here, the scale $\mathcal{B}$ of the spurious feature varies from $[-1, 1]$ (see Section C.3.1 for more details on this).

The way we construct the control and experimental dataset requires a bit more care here since we have 10 classes. Specifically, we replicate $S_{\text{inv}}$ ten times, and to each copy, we attach a spurious feature of a particular configuration. This creates $S_{\text{con}}$ which has no geometric or statistical skews. Also, the has a size of $|S_{\text{con}}| = 10|S_{\text{inv}}|$. Then, we consider the $1/10$th fraction of points in $S_{\text{con}}$ where the spurious feature has the correct configuration corresponding to the label. We create a large duplicate copy of this subset that is $81$ times larger than it (we do this by randomly sampling from that subset). We add this large duplicate set to $S_{\text{con}}$ to get $S_{\text{exp}}$. This gives rise to a dataset where for

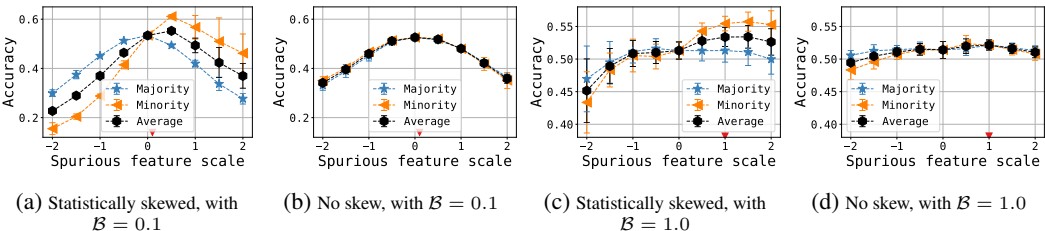

(a) Statistically skewed, with $\mathcal{B} = 0.1$

(b) No skew, with $\mathcal{B} = 0.1$

(c) Statistically skewed, with $\mathcal{B} = 1.0$

(d) No skew, with $\mathcal{B} = 1.0$

Figure 15: **Experiments validating the effect of statistical skews on the MNIST dataset in App C.4.1**. The red triangle denotes the value of $\mathcal{B}$ during training.

any label, there is a $10 : 1$ ratio[8] between whether the spurious feature matches the label or where it takes one of the other nine configurations.

**Observations.** We run experiments by setting $|S_{\text{inv}}| = 5k$ and so $|S_{\text{control}}| = 50k$ and $S_{\text{exp}} = 455k$. During training we try two different values of $\mathcal{B}$, $0.1$ and $1$ respectively. During testing, as in the previous section, we vary both the scale of the spurious feature and also its correlation by evaluating individually on the minority dataset (where the spurious features do not match the label) and majority datasets (where the spurious features match the label). Here, again we observe that the model trained on the non-skewed dataset is less robust, evidently due to the statistical skews.

It is worth noting that even though there is no statistical or geometric skew in the control dataset, we observe in Fig 15 (b) that the classifier is not completely robust to shifts in the spurious feature. We suspect that this may point to other causes of failure specific to how neural network models train and learn representations.

# D  SOLUTIONS TO THE OOD PROBLEM

Our discussion regarding geometric and statistical skews tells us about why ERM fails. A natural follow-up is to ask: how do we fix the effect of these skews to learn a good classifier? Below, we outline some natural solutions inspired by our insights.

**Solution for geometric skews.** Our goal is to learn a margin-based classifier that is not biased by geometric skews, and therefore avoids using the spurious feature. Recall that we have a majority subset $S_{\text{maj}}$ of the training set which corresponds to datapoints where $x_{\text{sp}} \cdot y > 0$ and a minority subset $S_{\text{min}}$ which corresponds to datapoints where $x_{\text{sp}} \cdot y < 0$. Our insight from the geometric skews setting is that the max-margin classifier fails because it is much "harder" (in terms of $\ell_2$ norm) to classify the majority dataset $S_{\text{maj}}$ using $\mathbf{x}_{\text{inv}}$ when compared to classifying $S_{\text{min}}$ using $\mathbf{x}_{\text{inv}}$. A natural way to counter this effect would be to somehow bring the difficulty levels of these datasets closer. In particular, we propose a "balanced" max-margin classifier $\mathbf{w}_{\text{bal}}$ that does the following:

$$\mathbf{w}_{\text{bal}} = \arg \max_{\|\mathbf{w}_{\text{bal}}\|=1} \min \left( \underbrace{\{\mathbf{w}_{\text{bal}} \cdot \mathbf{x} | \mathbf{x} \in S_{\text{maj}}\}}_{\text{margin on majority group}}, \quad \underbrace{\{c \cdot \mathbf{w}_{\text{bal}} \cdot \mathbf{x} | \mathbf{x} \in S_{\text{min}}\}}_{\text{downscaled margin on minority group}} \right)$$

where $c$ is a sufficiently small constant. In words, we consider a classifier that maximizes a "balanced" margin on the dataset. The balanced margin is computed by scaling down the margins on the minority datapoints, thereby making that subset artificially harder, and thus diminishing the geometric skew. For an appropriately small choice of $c$, we can expect the balanced max-margin to minimize its reliance on the spurious feature.

---

[8] Here's the calculation: in $S_{\text{control}}$, we have a subset of size $|S_{\text{inv}}|$ where the spurious feature is aligned with the label and in the remaining $9|S_{\text{inv}}|$ datapoint, the spurious feature is not aligned. So, if we add $81|S_{\text{inv}}|$ datapoints with matching spurious features, we'll have a a dataset where $90|S_{\text{inv}}|$ datapoints have matching spurious features while $9|S_{\text{inv}}|$ don't, thus creating the desired $10 : 1$ ratio.

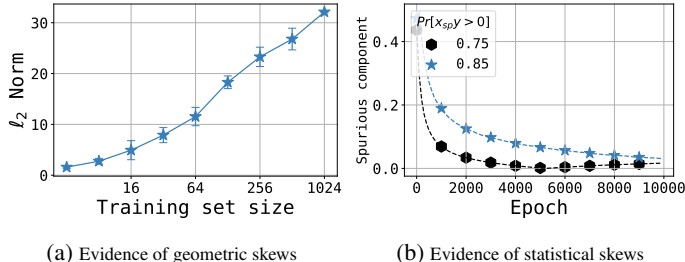

(a) Evidence of geometric skews       (b) Evidence of statistical skews

Figure 16: **Experiments validating geometric and statistical skews on the obesity dataset**: In Fig 16a, we show that the $\ell_2$ norm required to fit the data in the invariant feature space grows with dataset size. As discussed earlier, this would result in a geometric skew, which explains why a max-margin classifier would rely on the spuriously correlated feature. In Fig 16b, we show that the spurious component $w_{\text{sp}}/\|\mathbf{w}\|$ of the gradient descent trained classifier converges to zero very slowly depending on the level of statistical skew.[9]

Note that this sort of an algorithm is applicable only in settings like those in fairness literature where we know which subset of the data corresponds to the minority group and which subset corresponds to the majority group.

**Solution for statistical skews.** One natural way to minimize the effect of statistical skews is to use $\ell_2$ weight decay while learning the classifier: weight decay is known to exponentially speed up the convergence rate of gradient descent, leading to the max-margin solution in polynomial time. Note that this doesn't require any information about which datapoint belongs to the minority group and which to the majority.

In the setting where we do have such information, we can consider another solution: simply over-sample the minority dataset while running gradient descent. This would result in a dataset with no statistical skews, and should hence completely nullify the effect of statistical skews. Thus, our argument provides a theoretical justification for importance sampling for OoD generalization.

# E  DEMONSTRATING SKEWS ON A NON-IMAGE-CLASSIFICATION DATASET

So far, we have demonstrated our insights in the context of image classification tasks. However, our theoretical framework is abstract enough for us to apply these insights even in non-image classification tasks. To demonstrate this, we consider an obesity estimation task based on the dataset from Palechor & de la Hoz Manotas (2019).

**Dataset details.** The dataset consists of 16 features including height, weight, gender and habits of a person. The label takes one of six different values corresponding to varying levels of obesity. We convert this to a binary classification task by considering the first three levels as one class and the last three levels as another; we ignore any datapoint from the middle level, resulting in a dataset of 1892 points. We randomly split the data to extract 729 test datapoints. The dataset also has a categorical variable corresponding to the preferred mode of transport of the individual, which we convert to five binary features (corresponding to automobile, motorbike, bike, public transport and walking). We then scale all the resulting 20 features to lie between $-1$ and $1$. For all our experiments, we will consider fitting a linear classifier directly on these features.

Next, we construct a "biased" training set by sampling from the above training set in a way that the public transport feature becomes spuriously correlated with the label. For convenience we denote this feature as $x_{\text{sp}}$ and the remaining features as $\mathbf{x}_{\text{inv}}$. Note that in the original dataset there is not much correlation between $x_{sp}$ and $y$. In particular $\Pr_{\mathcal{D}_{\text{train}}}[x_s p \cdot y > 0] \approx 0.47$ (a value close to $0.5$ implies no spurious correlation). Indeed, a max-margin classifier $\mathbf{w}$ trained on the original dataset does *not* rely on this feature. In particular, $w_{\text{sp}}/\|\mathbf{w}\|$ is as small as $0.008$. Furthermore, this classifier achieves perfect accuracy on all of the test dataset. However, as we discuss below, the classifier learned on the biased set does rely on the spurious feature.

**Geometric skews.** We sample a biased training dataset that has $581$ datapoints for which $x_{\text{sp}} = y$ and $10$ datapoints for which $x_{\text{sp}} = -y$. We then train a max-margin classifier on this dataset, and observe that its spurious component $w_{\text{sp}}/\|\mathbf{w}\|$ is as large as $0.12$ (about $15$ times larger than the component of the max-margin on the original dataset). But more importantly, the accuracy of this model on the test dataset where $x_{\text{sp}} = y$ is $99.5\%$ while the accuracy on a test dataset where $x_{\text{sp}} = -y$ is only $57.20\%$. In other words, the classifier learned here relies on the spurious feature, and suffers from poor accuracy on datapoints where the spurious feature does not align with the label. Why does this happen? Our theory says that this must be because of the fact that as we increase the number of training datapoints, it requires greater and greater $\ell_2$ max-margin norm to fit the data using only the invariant feature space (and ignoring the spurious feature). Indeed, we verify that this increase does happen, in Fig 16a.

**Statistical skews.** To demonstrate the effect of statistical skews, we take a similar approach as in earlier sections. We consider a dataset where we have the same number of unique data in the majority group (where $x_{\text{sp}} = y$) and in the minority group (where $x_{\text{sp}} = -y$). However, the majority group contains many duplicates of its unique points, outnumbering the minority group. In particular, we consider two different datasets of $500$ datapoints each, and in one dataset, the majority group forms $0.75$ fraction of the data, and in the other it forms a $0.85$ fraction of the data. Note that in both these datasets, the max-margin classifier does not rely on the spurious feature since there's no geometric skew.

To demonstrate the effect of the statistical skew, we train a linear classifier to minimize the logistic loss using SGD with a learning rate of $0.01$ and batch size of $32$ for as many as $10k$ epochs (which is well beyond the number of epochs required to fit the dataset to zero error). We then verify in Fig 16b that gradient descent takes a long time to let the spurious component of the classifier get close to its final value which is close to zero. This convergence rate, as we saw in our earlier experiments, is slower when the statistical skew is more prominent.

**Important note.** We must caution the reader that this demonstration is merely intended to showcase that our theoretical insights can help understand the workings of a classifier in a practically important, non-image classification dataset. However, we make no recommendations about how to deal with such high-risk tasks in practice. Such tasks require the practitioner to make careful ethical and social considerations which are beyond the scope of this paper. Indeed, empirical benchmarks for evaluating OoD generalization (Gulrajani & Lopez-Paz, 2020) are largely based on low-risk image classification tasks as it provides a safe yet reasonable test-bed for developing algorithms.

**Remark on synthetic vs. natural spurious feature shifts.** It would be a valuable exercise to validate our insights on datasets with naturally-embedded spurious features. However, in order to test our insights, it is necessary to have datasets where one can explicitly quantify and manipulate the spurious features e.g., we need this power to be able to discard the spurious feature and examine the $\ell_2$ norms of the max-margin in the invariant feature space. Currently though, OoD research lacks such datasets: we either have MNIST-like datasets with synthetic but quantifiable spurious features, or realistic datasets like PACS (Asadi et al., 2019), VLCS (Fang et al., 2013) where it is not even clear what the spurious feature is. The lack of datasets with natural yet quantifiable spurious features is a gap that is beyond the scope of this paper, and is worth being bridged by the OoD community in the future.

