# OpenReview forum: "Understanding the failure modes of out-of-distribution generalization"
_ICLR.cc/2021/Conference — ICLR 2021 Poster_

### Official Review · AnonReviewer2 · 2020-10-27
**Interesting simple "test case" for analyzing distribution shift; Poorly presented**

**Rating:** 6
**Confidence:** 3

**Review:**

The paper studies generalization under distribution shift, and tries to answer the question: why do ERM-based classifiers learn to rely on "spurious" features? They present a class of distributions called "easy-to-learn" that rules out several explanations given in recent work and isolates the spurious correlation phenomenon in the simplest possible setting. Even on "easy-to-learn" distributions, linear models obtained from ERM use spurious features owing to either the dynamics of gradient descent trained on separable data (very slow convergence to the max-margin classifier) or a certain geometric skew in the data.

Pros:
- The paper address a question of great interest to the ICLR community: generalization under distribution shifts.
- The "easy-to-learn" set of instances seems like a reasonable test-case to understand robustness to distribution shifts before moving to more complex setting.
- The fact that spurious feature use can arise on the "easy-to-learn" instances suggests that previous explanations are underpowered in the sense that they're ruled out by this construction, but the pattern still arises.
- The "statistical" skew shows how the dynamics of gradient descent can introduce brittleness  to distribution shift, albeit in a limited setting.

Cons:
- This paper is written is a way that's confusing and frequently difficult to follow. Rather than focus on explicating the core phenomenon well and supporting it with clear theorems and well-justified experiments, the authors at times take a shotgun approach and mention numerous constructions (often in-line), references to empirical results, descriptions of experiments, that seem to obfuscate rather than clarify the core phenomena they're presenting. For instance, the "geometric" skew in Section 4 is never precisely explained.
- In a similar vein, the paper does not justify the experiments they run in light of the phenomena they seek to "intuitively understand." For instance, in Section 4/5, the paper presents a litany of experiments, without explaining how they fit into the "easy-to-learn" framework or relate to a phenomena described by the theorems. It's possible there's a connection, but it's not evident to this reader.
- The experiments focus on a variety of synthetic distribution shifts. While this may be useful in understanding the constructions in the paper, it's not evident to what extent these 'skews' and the 'spurious/invariant' distinction given here can explain the lack of robustness observed in practice on "real" or "natural" distribution shifts.

==================

Update after rebuttal:
Thank you to the authors for their detailed response. The clarification and updates in the geometric skews section and the more explicit justification and connections between the framework/theoretical results and the experiments improved readability and clarity. I also appreciate putting greater weight on the theoretical contributions and "easy-to-learn" task definitions, which provide a simple but non-trivial test case for robustness research. I'm increasing my score accordingly.

---

> ### Author Response · Authors · 2020-11-17
> **Detailed response: Our experiments are well-justified +  new experiments without synthetic spurious feature + updated presentation**
>
> **Justification for experiments:** We are sorry to hear that you found the presentation of the experiments confusing. We have tried to improve this in the updated version.
>
> However, it seems like there’s disagreement among the reviews e.g.,
> - AnonReviewer4: “experiments are carefully designed based on the constraints… well in support of their theoretical claims.”;
> - AnonReviewer3: “The experiments were thoughtful and well-designed…  results are presented effectively”
> - AnonReviewer1: “a fairly rigorous empirical analysis of geometrical and statistical skews on CIFAR and MNIST”,  "paper is fairly well written overall"
>
> As such, these experiments have a precise role: they provide concrete motivation behind the abstract theoretical ideas that follow them. Here’s the justification for each inline experiment:
> - **The MNIST + random ReLU features experiment in Sec 3**: This identifies a concrete linear task where max-margin fails even though the task is easy. Identifying this really easy task helps us in justifying our subsequent definition of the easy-to-learn tasks via the constraints. These constraints are abstract, and the reader may not appreciate why we make them without that concrete experiment beforehand.
> - **The “increasing norm” experiments in Sec 4**: This empirically validates a key property of real-world invariant features that we subsequently use to explain why max-margin fails. Without demonstrating this observation upfront in Sec 4, the reader wouldn’t be comfortable digesting the insights from Thm1 which would all be based on an unconfirmed hypothesis.
> - **The GD experiment on the 2D dataset in Sec 5**: This empirically identifies a new failure mode that cannot be explained by the previous story in Sec 4. This then leads to the theoretical question of “how can you explain this empirical observation?” and thus, the rest of that section.
>
> **Relation between NN experiments and easy-to-learn tasks**: We apologize for not making this connection clear enough. We clarify this below, and in more detail in the revised submission.
> - In Sec 4, we present four non-linear classification tasks trained by a neural network. The first two are easy-to-learn. The last two tasks are NOT easy-to-learn -- these crucially break a constraint, which is why failure happens. But we can still argue how such a failure can be explained via a geometric argument like Thm 1 in all these tasks. The purpose of the brief main paper discussion is to only let the reader know that the geometric insights are not specific to the linear theoretical examples. We defer the exact connection between these tasks and Thm 1 to the App C.
> - In Sec 5, we present two non-linear tasks trained by an NN. Both of these are easy-to-learn tasks. In both, we create a dataset with no geometric skew, but with some statistical skew. By showing that the resulting NN classifier uses the spurious feature, we demonstrate our hypothesis that failure can happen even without geometric skews, as long as the statistical skew exists.
>
> **Synthetic spurious features**: You’re right: empirically demonstrating these insights on  datasets with realistic spurious feature shifts would be very illuminating! To address this:
> - In App E, we have added experiments demonstrating the effect of these skews in an obesity estimation task. Here, instead of introducing an artificial spurious feature, we sample the data in a biased way so as to induce a spurious correlation between an existing feature and the label.
> - Having said that, we clarify why we think **this is beyond the scope of our paper.** Since we theoretically study OoD -- rather than show non-theoretical numerical improvements of a proposed algorithm -- we need datasets where it is easy to quantify and intervene on/modify the spurious feature (e.g., we need this to show how norms increase with training set size when we use only the invariant feature). **However, in none of the existing realistic OoD benchmarks (PACS, VLCS etc.,) is there a way to quantify the spurious features**. Indeed, in prior work like IRM, RiskEx etc., all “theory-verifying” experiments are limited to the synthetic image dataset, Colored MNIST. In fact, we’ve even gone beyond this dataset to propose a variety of unique quantifiable, synthetic datasets. We’ve also added a cats vs. dogs datasets with a color shift in the revised version. The lack of a realistic + quantifiable dataset is an important gap, beyond the scope of this paper, that is hopefully addressed by the OoD community.
>
> **A broad note**: We’d also like to clarify that the experimental part of the paper forms less than quarter of our contributions. Most of our contribution lies in providing a theoretical understanding of OoD failure covered by the points you’ve mentioned in your positives.
>
> We hope that with these clarifications and our steps to address your concerns regarding the experiments and experiment discussion help you reconsider the evaluation of this paper! Thank you!

---

> ### Author Response · Authors · 2020-11-17
> **Response summary: Our experiments are well-justified +  new experiments without synthetic spurious feature + updated presentation**
>
> Thank you for your helpful review and criticisms.
>
> **Main points in our response:** We understand you have concerns about the experiments, specifically about (1) their presentation and (2) connection to theory and (3) regarding the synthetic nature of the spurious features. In our uploaded revision, we have tried to address all your three concerns.
> - The **first concern** is that the experiments described in-line weren’t presented in a well-justified way. In particular, you noted that as a result of this, the geometric skews went under-explained.
>    - We’ve updated the geometric skews section to make it clearer.
>    - We argue why those experiments are as such crucial for our narrative.
>    - We also note that it seems like the other reviewers felt that the experiments were well-justified and motivated.
>    - Nevertheless, we have improved the discussion.
> - The **second concern** was that we didn’t justify how the neural network experiments in Sec 4 and 5 relate to easy-to-learn tasks. We clarify this and rewrote these parts.
> - The **last concern** is that the spurious features are synthetic.
>   - To address this, we’ve added experiments in App E on a real-world non-image dataset without an artificial spurious feature.
>   - But we also explain why this is beyond our scope since none of the realistic datasets that are used by the OoD community have quantifiable spurious features that are necessary to run our “theory-verifying” experiments.
> - You also noted that our statistical skews result was in a limited setting. We addressed this by **extending theorem 2 to a much more general setting!** Namely, this holds for any easy-to-learn task without geometric skews (and not just that 2D task).
> - **We wish to emphasize that our paper is largely theoretical** --- at least three-fourth of our contributions is theoretical, and we would appreciate if you take these contributions into account when evaluating the paper
>
>
> Finally, since we’ve done our best to address your concerns and updated the paper, we hope you would consider increasing your score.

---

### Official Review · AnonReviewer3 · 2020-10-29
**Intriguing work that yields potentially fundamental insights about out-of-distribution prediction errorrs**

**Rating:** 8
**Confidence:** 3

**Review:**

I stand by my initial review that this is a strong submission, and having read through the other reviews and author responses, I am raising my confidence level as well (I think I have a solid grasp of this work's potential import). I disagree with critiques of the paper's novelty and practicality -- I think it provides new insights into OOD problems with substantive theory (not common) and provides actionable insights to boot. Also, the the revised manuscript is much improved. I hope this gets accepted.

-----

This submission presents a rigorous analysis of a subset of ways in which machine learning models can fail when encountering out-of-distribution (OOD) samples (often referred to as train/test skew or as train/scoring skew in industry). As the paper notes, the topic has received a great deal of attention, particularly under other guises ("domain adaptation"). However, much of that attention has aimed at pragmatic or heuristic solutions (various tricks to design or learn "invariant" features), while our fundamental understanding of what goes wrong in OOD situations remains incomplete. This paper aims to fill those gaps in understanding by studying simplified settings, and asking the question: why does a statistical model learn to use features susceptible to shift ("spurious" features) when the task can be solved using only safe ("invariant") features. After formulating five constraints (guaranteed to hold true for easy-to-learn tasks), they go on to show that failures come in two flavors: geometric skew and statistical skew. They analyze and explain each in turn, while also providing illustrative empirical results.

I like this work a lot (though I am more lukewarm on the paper itself, see below), and barring discovery of a fatal flaw during the discussion, I would advocate with some enthusiasm for its inclusion in the conference. The paper's claims are stated at the bottom of page 2 as:
1. Careful design and articulation of "easy-to-learn" settings in which there are few, if any, unmeasured variables that could confound the findings (a weakness in previous work on this topic).
2. Identification of two (but not the only two) distinct types of OOD failures that occur even in easy-to-learn settings, in the form of necessary and sufficient data "skews."
3. Experimental evidence to illustrate and support the analyses from (2.), along with enlightening discussion.

I agree with the paper's claims, though I admit that I was not previously familiar with, e.g., Sagawa 2019 or Tsipras 2019, and so cannot confidently situate this work amongst related research. I also feel my understanding may still be somewhat superficial -- I buy its arguments but don't have a particularly strong intuition yet for the two flavors of skew (particularly in non-toy settings).

This work has a very strong scientific flavor (not always true of machine learning research): I would liken the restriction to carefully designed "easy-to-learn" settings to a well-designed laboratory experiment in which there are few, if any, unmeasured variables that could confound the findings. It is very elegant and satisfying to read and think about. I would anticipate that this paper will inspire a lot of follow-up work, in which other researchers adopt the "easy-to-learn" and "skew" framework and terminology and even utilize the specific experimental designs in this paper. After all, machine learning researchers love adopting intellectual frameworks and benchmarks that they can build upon rapidly.

The "easy-to-learn" constraints articulated in Section 3.1 are sensible and clearly stated, and I am unable to find fault in them thus far. I agree with this statement on page 5: "any algorithm for solving OoD generalization should at the least hope to solve these easy-to-learn tasks well."

The experiments were thoughtful and well-designed, and their results are presented effectively: each plot, it seems, illustrates a particular point or supports a specific argument in the paper. For example, I like how Figures 2 and 3 serve as visual summaries of the geometric and statistical skew sections, respectively. A reader (particularly a savvy one familiar with the relevant related work) could probably skip Sections 4 and 5 (three pages total!) and still get the high level idea simply by skimming the plots and reading the captions of those two figures.

The largest weakness I perceive concerns the clarity and accessibility of the writing: for example, the connection drawn in Section 4 to the work on norms in over-parameterized neural nets is very interesting, but I'm not sure the text fully succeeds in further connecting it to OOD settings. In particular, certain details of the ongoing discussion of majority and minority groups aren't entirely clear (to me, at least)...are minority group samples available during training, just in smaller number? In that case, what is the OOD "shift" -- the prevalence of the minority group at test time?

Likewise, I'm not sure I really connected with the takeaway in Section 5 -- is it that early in the optimization, the "spurious" weights get updated repeatedly by an amount proportional to the spurious correlation, and that it then takes a long time to undo these updates, if at all? The statistical skew section is definitely more abstract and perhaps a little harder to connect to practical settings, vs. geometric skew where the bridge is the previous work on "norms."

My recommendation is to accept this submission, and at the moment, I am willing to advocate for it. However, it is entirely possible I am missing (or misunderstanding) key details, and so I am eager to discuss with the other reviewers.

---

> ### Author Response · Authors · 2020-11-17
> **Detailed response: Thanks for your efforts to understand our work precisely and in detail.**
>
> First, we want to address your final two questions.
>
> **Geometric skews**: We apologize for the clarity issues in the discussion of the majority/minority groups. We have rewritten this paragraph, but nevertheless clarify this here too.
>
> The majority and minority groups is a partition of the training data defined for the purpose of understanding how the max-margin classifier works. The max-margin classifier as such does not know which is which! How does the “increasing norms” observation explain the spurious-feature-reliance? There are four logical steps in this argument:
>  1. Note the (trivial) fact the number of points of the minority group $S_{\text{minor}}$ (defined by points for which $x_{\text{sp}} \cdot y < 0$) is (much) less than the size of the whole dataset $S$.
>  2. The “increasing norms” observation is that “if we train a max-margin classifier to fit the data using only the invariant features, its norm grows with number of training points.”
>  3. The “increasing norms” observation together with step 1 tells us that if we were to use only the invariant feature, fitting just $S_{\text{minor}}$ is (much) cheaper in $\ell_2$ norm than fitting all of $S$. This gap in costs is what we call a geometric skew.
>  4. Hence, the max-margin classifier would rather just set $w_{\text{sp}} > 0$ to classify $S_{\text{major}}$ and use the invariant feature to focus on classifying the remaining set, $S_{\text{minor}}$. This would be cheaper than setting $w_{\text{sp}} = 0$ and using the invariant feature to classify all of $S$. Thus, the max-margin uses the spurious feature because of the geometric skew.
>
>
> **Statistical skews:** What you’ve stated is precisely the takeaway i.e., gradient descent initially updates the spurious feature proportional to the spurious correlation, but it takes a long time to “correct” this back to a zero. We want to make two additional points:
> - We have updated Theorem 2 to apply to a more general easy-to-learn setting that doesn’t just apply to a 2-dimensional task. We hope this helps in convincing you that the above intuition is indeed more generally true.
> - The experiments in Sec 5 are intended to bridge these insights to practice. We’re able to show that a NN trained on an MNIST/CIFAR10-based dataset with statistical skews (but no geometric skews) is vulnerable to shifts in the spurious feature. This shows that the NN has not converged to a “max-margin-like” state that doesn’t use the spurious feature.
>
>
> **Regarding your (accurate) understanding:**  We believe that your understanding of our results, and also the background context is on point. Indeed, you make a good point that most existing algorithms in OoD in deep learning are often based on heuristics or on rough intuition.
>
> With regards to your point about related research, we think it’s useful to add that there has been very little theory understanding the exact factors behind why existing algorithms don’t work for OoD generalization. The ones we’ve cited like Sagawa et al., are very recent. Most other work in this space has been empirical.
>
> Your understanding regarding the motivation behind trying to study easy-to-learn tasks is accurate. This motivation is a very nuanced idea, and we’re pleased that you appreciated it. Indeed, as you say, we want to “switch off” confounding factors to tease out the most fundamental factors of failure.
>
> Thank you for finding our discussion elegant. We think that what is particularly appealing  about the result is in how these two skews are “orthogonal” effects. We agree with you that future work -- both theory and practice -- can hopefully build on these ideas.
>
> We’re also glad that you found the figures useful and the experiments well-justified. You’re right in that we made sure that Fig 2 conveyed the idea of geometric skews and Fig 3 the idea of statistical skews.

---

> > ### Comment · AnonReviewer3 · 2020-11-23
> > **Quick touch base**
> >
> > Authors,
> >
> > I want to commend you on your responses. You've done a nice job of acknowledging constructive criticism (especially regarding clarity) and answering questions while also advocating for your hard work, which you should.
> >
> > I've skimmed through the other reviews and your responses. I don't see anything to change my own favorable disposition toward your manuscript.
> >
> > Good luck!

---

> ### Author Response · Authors · 2020-11-17
> **Response summary: Thanks for your efforts to understand our work precisely and in detail.**
>
> We are delighted to read your positive and very elaborate review of the paper. Thank you!
>
> **Main points in our response:**
> You raised two questions about the paper, specifically regarding the geometric skews and statistical skews.
> - **Question 1 (geometric skews):** We’ve uploaded a revised version explaining the geometric skews more clearly especially by relating it to Figure 2(c).
> - **Question 2 (statistical skews):** Your understanding of the statistical skews is correct!
> - Relatedly, we’ve extended Theorem 2 (statistical skews) to a more general setting, confirming that these insights are more general!
> - Your understanding of the rest of the paper especially regarding the motivation behind the “easy-to-learn” experiments is accurate.
>
> In general, we enjoyed your elaborate and engaging feedback, indicating the depth of your understanding which is unfortunately (for us) not reflected in your confidence score.

---

### Official Review · AnonReviewer4 · 2020-10-29
**A good theoretical attempt to explain the failure of OOD generalization**

**Rating:** 6
**Confidence:** 4

**Review:**

This paper investigates the reasons why machine learning models usually fail to generalize out-of-distribution even in easy-to-learn tasks where one would expect these models to succeed. The authors propose two kinds of skews in the data: geometric and statistical, to explain this behavior and theoretically demonstrate it in the linear and easy-to-learn settings.

Pros:
+ The problem studied in this paper has been one of the most important in the community and this work has a meaningful attempt on the theoretical side.
+ The theoretical results obtained by analyzing the simplest tasks are insightful and could be seemingly drawn on in more general settings.
+ The experiments are carefully designed based on the constraints, which is well in support of their theoretical claims.

Major concerns:

I agree with the authors on most of their explanations on the failure modes of OOD generalization from the geometric and statistical perspectives in easy-to-learn tasks. However, the authors claim that these two skews are “not just a sufficient but also a necessary factor for failure of these models in easy-to-learn tasks“, which I think might be overclaimed, even in the easy-to-learn tasks defined in the paper. I do believe that both geometric and statistical skews play a role in the failure of OOD generalization, but they might not be the only reasons. In fact, even if there exist no spurious features, machine learning models might fail to generalize out-of-distribution as well.

Here is the example. Let us consider a comparison task, where we aim to learn a linear classifier to predict which one of any given two values {x_i, x_j} is larger, i.e., it outputs +1 if x_i >= x_j, and otherwise it outputs -1. For simplicity, let us further assume that -1 =< x_i, x_j =< +1 and x_i^2 + x_j^2 =1. In this case, x_inv = (x_i, x_j) and there is no x_sp. It is also easy to check that this task satisfies the five constraints stated in the paper. Given the training dataset in which all data points (x_i, x_j) are satisfying that 0 =< x_i, x_j =< +1 and x_i^2 + x_j^2 =1, it is easy to learn a linear classifier that can be well generalized to the whole domain [-1, +1]. If we represent  (x_i, x_j) in the polar coordinate system, then (x_i, x_j) is converted to (1, \theta). In this case, we have x_inv = \theta and no x_sp, which also satisfies the five constraints. Although we still can learn a perfect linear classifier in the training data, it is impossible to generalize to the whole [-\pi, +\pi] corresponding to [-1, +1]. The reason is that this task is no longer linearly separable. From the example above, we can see that even without spurious features, some simple transformations only on invariant features would render OOD generalization impossible. This would be another failure mode beyond the geometric and statistical skews.

Minor concerns:
- The constraint on fully predictive invariant features seemingly simplifies the OOD generalization problem too much, making the theoretical results hard to apply to more general settings. Actually this constraint circumvents two key questions which play a pivotal role in the OOD generalization. One is how to justify whether or not features are fully predictive and invariant across all the domains or environments. This involves the generalization assumption, like Assumption 8 of Arjovsky et al. (2019). The other is how to learn the partially predictive invariant features which, albeit not fully predictive, remains invariant across all the domains. If we want to solve the OOD generalization problem in practice, these two questions have to be answered.

- The example given in Constraint 3.2 is not suitable. The distribution of invariant features must be identical across all domains, otherwise these features should not be treated as “invariant”. The exact distribution of the shapes of cows and camels vary across domains because they are not the true invariant features.

- Constraint 3.5 is too strong as well. It circumvents another key question in the OOD generalization: how to identify x_inv and x_sp. If we can identify them, the OOD generalization would be reduced to a simple ERM problem.

Other comments:
- The last 6th line before section 3.1 says that Pr_{D_test}[x_sp \cdot y >0] = 0.0. I believe this is a typo, right?
- Overall I feel too much content is placed in the appendix and it affects the fluency of reading a bit. I suggest that the authors re-organize the content and put back some stuff to the main text.

---

> ### Author Response · Authors · 2020-11-17
> **Detailed response: Why the example isn’t a counter-example + clarifying all minor concerns**
>
> **Major concern:** We understand that your point is that there is a counter-example to our claim that “geometric/statistical skews are necessary for failure in easy-to-learn tasks”. However, we believe that this is not a counter-example as it is not an easy-to learn task: it breaks Constraint 2 i.e., the invariant feature distribution must be the same in training and testing. This is not obeyed in your example since test-time points have new invariant feature values, thus making your task particularly harder than the easy-to-learn tasks!
>
> Also, it’s worth making a minor clarification: our setting considers the realizable case where the optimal hypothesis $h^*$  in the learner’s hypothesis class $H$ (defined under the “the domain generalization setting and ERM” paragraph) achieves zero error (captured by Constraint 1). In your example, the optimal hypothesis from the ERM learner’s linear hypothesis cannot achieve zero error thus violating Constraint 1 too. So your task is hard-to-learn in the sense that it is in the non-realizable setting and violates Constraint 1. *Nevertheless, the main issue with the example is that it breaks Constraint 2.*
>
> Finally, we’d like pre-emptively clarify that our “necessary and sufficient” claim is intended only for (a) easy-to-learn tasks and (b) for linear, GD/max-margin. This claim is a consequence of the upper and lower bounds in Thm 1 and 2. We had made sure to phrase this carefully in Contribution 3.
>
> Having said all this, we want to thank you again for engaging with the material deeply.
>
> **Minor Concern 1:** Your concern suggests that the set of easy-to-learn tasks doesn’t contain complicated real-world tasks where the invariant feature isn’t fully predictive. While we completely agree with this fact (and we’ve admitted this ourselves in the conclusion), we must remind ourselves of a fundamental nuance here (one which you’ve noted yourself in your paper summary!). The nuance is that **our goal here is not to prove the success of a new OoD algorithm** -- had it been that, your concern regarding the fully predictive invariant feature is valid! OTOH, we want to show that ERM fails. **It’s more powerful (and most challenging) to show that an algorithm fails to work in a task that is easy where the invariant features are fully predictive, than one that is hard**. Hence this is not a problematic aspect but in contrast, a highlight of our result!
>
> Further, we think it’s natural that the insights and failure modes we elucidate here carry over to more complicated tasks where the invariant feature may not be fully predictive. For a concrete example: GD would still be slow in its convergence on more complicated tasks (where invariant feature isn’t fully predictive), and hence still suffer from statistical skews (as it’ll absorb the spurious correlation early on, and take too long to get rid of it). Additionally, we think our experiments on neural networks + CIFAR-like datasets are concrete evidence for the generality of our insights.
>
> **Minor Concern 2**: Here’s why the example *is* suitable. Consider an example with two types of cow faces (A & B) and two types of camel faces (C & D). In the first domain all four types are equally probable, and in the second domain, A and C are most probable. This would break Constraint 2 since the shape feature varies in distribution across domains. Yet, the “shape feature → label” mapping remains the same in both domains, and so the shape feature can be considered an invariant feature that fully predicts the true label.
>
>
> **Minor Concern 3:** To clarify, this constraint did not mean to imply that the learner knows which feature is spurious and which feature is invariant. More precisely, both our Theorems hold in the case where the original feature space is orthogonally and arbitrarily rotated (and the orthogonal rotation matrix is not known to the learner) -- this holds because we consider GD/max-margin which are equivariant to such transformations. We wrote the constraint in this way just for ease of notation but we understand the confusion and apologize for it. We have clarified this in our updated paper.
>
> **Your other comments:**
> - We’re not sure we see a typo here --- could you clarify what you think it should have been? During test time, we want $x_{\text{sp}} \cdot y < 0$ on all datapoints in contrast to training time where $x_{\text{sp}} \cdot y > 0$ on most datapoints.
> - In other words, we want to “flip” the correlation completely to test the robustness of the learned classifier.
> Regarding appendix references, we’ve tried to reorganize/add some content especially in the discussion of geometric skews, and in the empirical examples in Sec 4. We’ll continue working on the re-organization. Thanks for the useful feedback.
>
>
> To summarize, we are hopeful that we have addressed your major concern and provided the required clarifications for your minor concerns too. If that is the case, we appreciate it if you consider increasing your score.

---

> ### Author Response · Authors · 2020-11-17
> **Response summary: Thanks for thoughtful review! Regarding your example: it’s a nice one but not a counter-example.**
>
> **Key points of our response:** We’re excited to see that you’ve written a positive review, and have carefully engaged with our ideas to come up with a well-thought critique! Thank you! Briefly, here are our clarifications addressing all your concerns:
>
> **Major concern:** Your major concern is regarding our claim that the skews are necessary for failure. You’ve a nice example, but we argue that it is not a counter-example as it’s not an easy-to-learn task. It breaks constraint 2 (and also constraint 1).
>
> Minor concerns:
> - Concern 1 is that the fully predictive invariant features constraint simplifies the OoD problem too much: Our goal is not to find a solution to OoD, but to show failure of ERM! It’s more powerful to show that ERM fails in an easier setting with a fully predictive invariant feature!
>
> The next two concerns are simple misunderstandings stemming from our phrasing. We apologize and have fixed the phrasing:
> - Concern 2 is that the quick example we give to illustrate Constraint 2 isn’t suitable. We clarify why this is suitable: the shapes of cows/camels can be an invariant feature while varying in distribution since these features can still fully predict the label.
> - Concern 3 is that Constraint 5 seems to “reveal” the identity of the spurious feature. We clarify that our theory holds even if you “obscure” the feature’s identity by orthogonally rotating the data in an arbitrary way (unknown to the learner). We defined the last feature to be $x_{\text{sp}}$ just for ease of exposition. We have clarified this in our updated paper.
>
> **Sidenote:** we’ve made Theorem 2 to apply to a general easy-to-learn task!
>
> Since we think we have addressed your major concern and your three minor concerns (and have updated the paper to clarify these), we hope you would consider increasing your score. Thank you!

---

### Official Review · AnonReviewer1 · 2020-10-30
**a fair paper but lacking in novelty**

**Rating:** 5
**Confidence:** 3

**Review:**

Overview: The authors study the out of distribution generalisation problem in detail wherein a model may incorrectly use spurious correlations in the data to make predictions on data at test time. This is a fundamental problem and very crucial/difficult to tackle across various domains

Quality and Clarity: The paper is fairly well written overall.

Originality and Significance: OOD detection is a well-studied problem that has proven difficult to date; it's highly relevant across applications in high-stake domains. The approach taken in this paper is to a) study tasks that are easy to succeed on, b) show that OOD failure occurs even in these easy settings. The authors subsequently show that geometric and statistical skews are necessary for failure. As such the approach has limited novelty, since it does not offer a concrete solution to the OOD detection problem but rather examines necessary conditions for OOD failure.

Pros: 1) The authors study a highly relevant problem of OOD failures
2) The authors present a fairly rigorous empirical analysis of geometrical and statistical skews on CIFAR and MNIST

Cons: 1) The paper is limited in novelty; while the authors state find that OOD failures result from statistical and geometric skews, they do not provide an adequate solution to overcoming these skews

2) OOD failures are only studied in the context of small theoretical examples or image tasks such as MNIST or CIFAR. However, OOD failures are widespread across various domains and become especially important to study in high risk settings such as in a clinical context etc. I would have liked to have seen some exposition of these problems in a real high-risk setting.

---

> ### Author Response · Authors · 2020-11-17
> **Detailed response (part 2): There are **many** deep learning theory papers that solely understand an empirical phenomenon without suggesting solutions**
>
> Finally, we think it would be helpful to place our paper in the context of current trends in ML theory. In particular, there is a large swathe of theoretical papers these days that are solely dedicated towards understanding a particular empirically observed phenomenon (such as the success/failure of an existing deep learning algorithm) -- **these papers do not provide a new solution.** We present a few examples at the end of this comment.
>
> Such papers (including ours, we hope) are valuable in themselves as they make novel points, provide stronger theoretical foundations, add rigor to the field, generate significant discussion and introspection on empirical progress. We feel that penalizing such work for not immediately solving the studied phenomenon would cut short a promising line of future work. We hope that keeping this in mind provides a broader perspective while approaching the question of what makes our contributions a standalone paper.
>
> Examples of works that solely understand an existing algorithm’s working/failure **without proposing any new algorithms:**
>
> - An investigation of why overparameterization exacerbates spurious correlations, ICML 2020 https://arxiv.org/abs/2005.04345 (a study of why overparameterized max-margin makes the effect of spurious correlations worse -- we cite this)
> - Pitfalls of simplicity bias in neural networks, NeurIPS 2020 https://arxiv.org/abs/2006.07710 (a study of how standard NN training ends up learning to rely on features that separate the data only by a small margin just because they are simple to learn)
> - Robustness may be at odds with accuracy,  ICLR 2019
>  https://arxiv.org/abs/1805.12152 (a study of why robust training could hurt standard accuracy)
> - Adversarially Robust Generalization Requires More Data, NeurIPS 2018 https://arxiv.org/abs/1804.11285 (a study explaining why robust training fails to generalize to unseen data)
> - Identifying and understanding empirical phenomena, ICML 2019 workshop http://deep-phenomena.org/ (a workshop dedicated to works like this)
> - The risks of invariant risk minimization, https://arxiv.org/abs/2010.05761 (very recent work studying the pitfalls of IRM)

---

> ### Author Response · Authors · 2020-11-17
> **Detailed response (part 1):  Why our insights in themselves are a novel and important contribution + why high risk datasets are beyond our scope**
>
> **Why our insights are novel, general and significant:** We agree that providing solutions to OoD generalization is a valuable (in fact, the ultimate) goal. Our paper intends to provide a strong theoretical foundation that is **currently lacking** to make progress towards that very goal. As pointed out by AnonReviewer3, OoD algorithms are based upon heuristics or handwavy intuition, which makes it hard to know when these may fail, and how to improve them soundly. Moreover, there has been little theory so far (two or three papers) that have tried to understand why ERM uses spurious correlations.
>
> We show that this behavior happens even in dead easy tasks which is surprising and counter-intuitive (Section 3 last para) and has neither been identified formally nor explained so far. Understanding why ERM fails even in such easy tasks is powerful as it uncovers **previously-unknown fundamental flaws of ERM** (rather than some superficial flaws). This opens up the possibility for future work to address these fundamental flaws and thereby find a more general solution rather than find superficial fixes.
>
> Furthermore, the tasks we study are general -- e.g., we make little assumption about what the invariant features should look like -- when compared to all **prior theory work which study only concrete settings**. We also want to point out that we’ve updated Theorem 2 to be a lot more general. As opined by other reviewers, we believe our insights are general enough and hopefully will result in a lot of follow-up work:
> - AnonReviewer3: "this paper will inspire a lot of follow-up work, in which other researchers adopt the "easy-to-learn" and "skew" framework and terminology and even utilize the specific experimental designs.” and
> - AnonReviewer4: “The theoretical results obtained by analyzing the simplest tasks are insightful and could be seemingly drawn on in more general settings”,
>
> **Concrete solutions:** To demonstrate the usefulness of our insights as you requested, in App D, we propose concrete solutions to fix both geometric and statistical skews. These ideas are natural, indicating that identifying how these failure modes arise constitutes the hardest part of the challenge. In short:
> - for statistical skews, we propose applying weight decay as it leads to faster convergence to the max-margin.
> - for statistical skews, we could also simply “oversample” the minority group.
> - for geometric skews, we propose a “balanced max-margin” classifier by scaling down the margins on the minority dataset. We argue this brings down the geometric skew and hence would force the max-margin to ignore the spurious feature.
>
> **High-risk & practically important datasets:** Certainly, this would be a valuable addition to OoD research. As you requested, in App E, we’ve addressed this by studying an obesity estimation dataset which is (a) not only an non-image classification task but (b) also doesn’t have synthetically introduced spurious features. We induce a spurious correlation simply via sampling bias. We demonstrate both geometric and statistical skews in this non-image classification task.
>
> Nevertheless, we feel it’s important to also clarify why **this is beyond our scope**:
> - **Benchmark datasets for OoD algorithms are image datasets** (Gulrajani & Lopez-Paz, ‘20). Hence, this criticism applies in general to the field and less so to a theoretical paper like ours that tries to understand failure rather than make numerical improvements in the solution. We think that exploring high-risk datasets is an important gap to be addressed by the OoD community at large, but beyond the scope of a theoretical paper.
> - At the same time, high-risk datasets entail significant ethical considerations. We feel it’s wise for an ML theory paper to keep the discussion abstract enough while concretely demonstrating them in non-risky examples, which we do.
>
> We strongly hope that the new results we’ve added to address your concerns, together with our above arguments help you view our paper in a broader light and help appreciate the true significance of such work in bolstering future empirical and algorithmic work.

---

> ### Author Response · Authors · 2020-11-17
> **Response summary:  Added concrete solutions and new experiments on non-image classification; but we think this is beyond our scope**
>
> **Key points in our response:** Thank you for taking the time to review our work. We understand you have two concerns which is that you’d like to (1) see demonstrations of our skews on practically-relevant, high-risk datasets and (2) find solutions to OoD based on these insights. While we agree this’d be exciting, we also think this is beyond the scope of our paper as we explain later. But since these are your major concerns, we’ve addressed both completely in the updated paper as follows:
> - **Reg. Concern 1:** In App. D, we propose natural solutions to overcome both geometric and statistical skews that are inspired by our failure modes.
> - **Reg. Concern 2:** In App. E, we study a non-image, obesity estimation task, demonstrating both geometric and statistical skews! (But we want to add the caveat that we’re not domain experts in such sensitive, socially related and high-risk settings and this is done only to demonstrate that our understanding can help in other domains.)
>
> Having said that, here’s why we think **this is all beyond the scope of our paper:**
> - Identifying and rigorously formulating these failures that occur even in the easiest of tasks is in itself challenging.  By doing this, our work provides a theoretical foundation that is **lacking** in current OoD research.
> - We also point out (in a later comment) a rich space of theory papers that merely explain an empirical phenomenon without proposing solutions and such papers are in themselves valuable.
> - Existing empirical OoD research has itself been largely limited to low-risk image classification tasks.
>
> Thus, we sincerely hope that you’d agree with us as to why our results as a standalone make a novel and valuable paper. On top of that, since we’ve also addressed both your concerns, we hope you’ll consider increasing the score to “accept”. Thank you!

---

### Decision · Program_Chairs · 2021-01-07
**Final Decision**

**Decision:**

Accept (Poster)

**Comment:**

This paper studies the reasons for failure of trained neural network models on out of distribution tasks. While the reviewers liked the theoretical aspects of the paper, one important concern is about the applicability of these insights to real datasets. The authors added an appendix to the paper showing results on a real dataset that mitigates this concern to an extent. Further, there are interesting insights in the paper to merit acceptance.